# PERSONALIZED FEDERATED LEARNING VIA TAILORED LORENTZ SPACE

## ABSTRACT

Personalized Federated Learning (PFL) has gained attention for privacy-preserving training on heterogeneous data. However, existing methods fail to capture the unique inherent geometric properties across diverse datasets by assuming a unified Euclidean space for all data distributions. Drawing on hyperbolic geometry's ability to fit complex data properties, we present FlatLand[1], a novel personalized **F**ederated **lea**rning method that embeds different clients' data in **t**ailored **L**orentz space. FlatLand is able to directly tackle the challenge of heterogeneity through the personalized curvatures of their respective Lorentz model of hyperbolic geometry, which is manifested by the time-like dimension. Leveraging the Lorentz model properties, we further design a parameter decoupling strategy that enables direct server aggregation of common client information, with reduced heterogeneity interference and without the need for client-wise similarity estimation. To the best of our knowledge, this is the first attempt to incorporate hyperbolic geometry into personalized federated learning. Empirical results on various federated graph learning tasks demonstrate that FlatLand achieves superior performance, particularly in low-dimensional settings.

## 1 INTRODUCTION

Federated learning (FL) trains machine learning models across multiple clients while ensuring data privacy. Traditional FL struggles with data heterogeneity, as one model cannot satisfy diverse local requirements. Personalized federated learning (PFL) resolves this by sharing common model knowledge and allowing for client-specific adaptations. PFL approaches mainly address heterogeneity through three strategies during aggregation: (1) splitting models into shared and personalized components (McMahan et al., 2017; Tan et al., 2023); (2) analyzing weights/gradients to evaluate client similarities (Xie et al., 2021); or (3) incorporating additional modules to enable client-specific customization (Baek et al., 2023). All these methods are conducted in Euclidean space.

Recent studies in various domains, including text (Tifrea et al., 2018; Dhingra et al., 2018), images (Atigh et al., 2022; Khrulkov et al., 2020), and graphs (Chami et al., 2019; Tan et al., 2023; Yang et al., 2022b;a), have shown that real-world data exhibit non-Euclidean properties, such as scale-free structures and implicit hierarchical relationships (Albert & Barabási, 2002; Khrulkov et al., 2020). Euclidean space, being inherently "flat", fails to adequately represent these characteristics, leading to structural distortions and reduced performance (Chami et al., 2019). For example, the CiteSeer graph dataset partitioned into 10 clients, shows varying degree distributions with long-tail characteristics which are poorly captured by Euclidean geometry, as illustrated in Figure 1(a). Besides, we calculate the Ricci curvature values of multiple real-world graph datasets after splitting them into 10 clients each and observe that they all exhibit negative Ricci curvature with significantly varying values, as shown in Figure 6. Higher absolute values indicate more pronounced non-Euclidean properties.

Moreover, embedding data from various clients into a fixed Euclidean space complicates interpretability of model parameters. All parameters play the same role during training, obscuring which encapsulates client heterogeneity versus shared information. This makes it difficult to segment the model into meaningful components and assess client similarity. Additionally, incorporating extra modules to aid this process escalates complexity and reduces flexibility.

---

[1]Our method is named after Edwin Abbott's book "*Flatland: A Romance of Many Dimensions*", highlighting our insights of exploring an extra dimension that maps various data distributions onto different Lorentz surfaces.

The aforementioned problems inspire us to ask **whether there is a space where we can design a tailored model for each client, in which we can *effectively* represent the inherent properties of local data and *succinctly* reflect the heterogeneity without any extra calculations?**

We propose to leverage **Lorentz Space**. With negative curvature, Lorentz space has the advantage of modeling complex data, particularly hierarchical, tree-like, and power-law distributed data (Lensink et al., 2022; Dhingra et al., 2018; Sun et al., 2022). By adjusting its curvature, it offers personalized and precise data representations for each client, leveraging its unique time-like dimension to capture diversity. This inspires us to design a framework that embeds

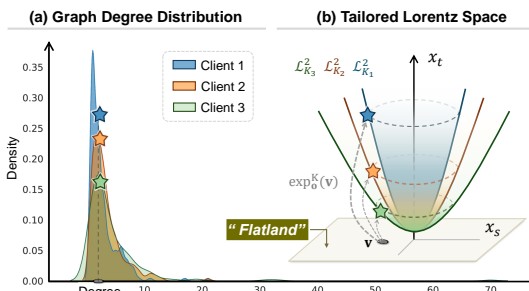

Figure 1: Toy example: (a) KDE of degree distributions from three CiteSeer clients (Davis et al., 2011), and (b) their respective 2D Lorentz Spaces with different curvatures $K$.

each client's data into a suitable Lorentz space. This will bridge the gap between the fields of hyperbolic geometry and personalized federated learning.

Furthermore, the representations in Lorentz space and the operations of Lorentz neural networks (Chen et al., 2021) have stronger interpretability. Take Figure 1(b) as an example[2]. Informally speaking, the diversity of the distribution can be more prominently represented by the *"height"* of the additional *time-like* dimension ($x_t \in \mathbb{R}$) while maintaining the relatively similar properties in the *"Flatland"* (*space-like* dimensions $\mathbf{x}_s \in \mathbb{R}^d$). In this work, we focus on federated graph learning (FGL) as hyperbolic encoders have achieved state-of-the-art results in many benchmarks (Atigh et al., 2022; Peng et al., 2021; Lensink et al., 2022). And there is a theoretical guarantee connecting the heterogeneity of graph data with hyperbolic curvature (Krioukov et al., 2010). This method is generalizable to other datasets and settings.

Although the Lorentz space has demonstrated significant potential in various tasks (Peng et al., 2021; Atigh et al., 2022), applying it to personalized federated learning (PFL) scenarios is still non-trivial. The challenge is **how to mitigate the influence of parameters related to heterogeneous information**, and aggregate the parameters that represent common features in the *"Flatland"* without accessing client data?

Motivated by the above insights, we propose an exploratory personalized **F**ederated **lea**rning method that embeds different clients' data in **T**ailored **L**orentz space, called FlatLand. To address the challenge, we formulate a **novel parameter decoupling strategy** that can directly aggregate shared parameters without any extra similarity calculations.

To the best of our knowledge, FlatLand is the first work to incorporate Lorentz geometry into personalized federated learning. It is **succinct**, **effective**, and **easily interpretable**. Experimental results demonstrate that FlatLand achieves superior performance than its Euclidean counterpart, particularly in low-dimensional representations.

## 2 RELATED WORK

**Personalized Federated Learning** With statistical heterogeneity (Kairouz et al., 2021), conventional FL frameworks like FedAvg (McMahan et al., 2017) can hardly obtain a single global model that generalizes well to every client (the basic framework is shown in Appendix A.4). Motivated by this, researchers have proposed personalized FL (PFL) to train customized local models. Generally speaking, existing PFL techniques can be categorized into the following three groups: (1) techniques that personalize client models via local fine-tuning (Fallah et al., 2020; Jiang et al., 2019; Wang et al., 2019), (2) techniques that personalize client models via customized model aggregation (Huang et al., 2021; Li et al., 2021b; Luo & Wu, 2022; Sun et al., 2021; Zhang et al., 2023b; 2021b), and (3) techniques that personalize client models via creating localized models/layers (Arivazhagan et al., 2019; Chen & Chao, 2022; Collins et al., 2021; Deng et al., 2020; Dinh et al., 2020; Hanzely &

---

[2]For convenience, all origins of Lorentz spaces in the figure are shown as the same, but actually, their origins are not in the same location.

Richtárik, 2020; Li et al., 2021a; Mansour et al., 2020). However, these PFL methods typically operate in Euclidean spaces to encode data samples, which can hardly capture the scale-free property and implicit hierarchical structure embedded within client data.

**Personalized Federated Graph Learning**  When applied to graph data, personalized federated graph learning (PFGL) can intuitively exhibit the problem mentioned above. For example, Xie et al. (2021) clusters clients based on gradients to aggregate models with similar data distributions. Another method (Tan et al., 2023) introduces additional personalized models to capture client-specific knowledge of graph structure. Baek et al. (2023) calculates client-client similarities to apply personalized model aggregation with local weight masking. All these methods learn node representations in Euclidean spaces, which cannot model the power-law degree distributions that widely exist in real-world graph data (Albert & Barabási, 2002; Krioukov et al., 2010). Additionally, the client clustering procedure and additional model components introduce computational overhead that may not be feasible in real-world scenarios with strict privacy constraints or limited resources.

**Hyperbolic Federated Learning**  Very few research works have considered incorporating hyperbolic spaces into federated settings. An et al. (2024) leverages hyperbolic distances to distill knowledge from the global model to the local model, to mitigate model inconsistency caused by data heterogeneity. Liao et al. (2023) applies hyperbolic prototype learning to capture the hierarchical structure among data samples. As the work most similar to our FlatLand, FedHGCN (Du et al., 2024) is a simple combination of FedAvg and hyperbolic graph neural networks along with a node selection process. Although these methods can benefit from the hyperbolic space to capture the hierarchical structure in the data, they do not have the personalization capability to adaptively model client data spaces with different curvatures. This may lead to suboptimal results when there is severe data heterogeneity. Therefore, our goal is to design a novel FL framework that can encode client data in hyperbolic spaces with adaptive curvatures using personalization techniques.

## 3 PRELIMINARIES

**Lorentz Manifold**  Given a $d$-dimensional Lorentz manifold $\mathcal{L}_K^d$ with a constant negative curvature $-1/K(K > 0)$, suppose a point / vector $\mathbf{x} \in \mathcal{L}_K^d$, which has the form $\mathbf{x} = \begin{bmatrix} x_t \\ \mathbf{x}_s \end{bmatrix} \in \mathbb{R}^{d+1}$, where the first dimension $x_t \in \mathbb{R}$ is called *time-like* dimension and others $\mathbf{x}_s \in \mathbb{R}^d$ are *space-like* dimensions. It satisfies the following conditions: $\langle \mathbf{x}, \mathbf{x} \rangle_{\mathcal{L}} = -K$ and $x_t > 0$, where $\langle \mathbf{x}, \mathbf{y} \rangle_{\mathcal{L}} = -x_t y_t + \mathbf{x}_s^\top \mathbf{y}_s$ is the Lorentzian inner product. Note that the larger the $K$, the more the intrinsic structure of the data deviates from the flatness of Euclidean space. Formal definitions are shown in Appendix A.1.

Typically, inputs reside in Euclidean space and need to be mapped into hyperbolic space. The way of projecting the data $\mathbf{v}^E \in \mathbb{R}^d$ in Euclidean to Lorentz space $\mathbf{x} \in \mathcal{L}_K^d$ can be simplified as [3]

$$\mathbf{x}^K = \exp_{\mathbf{o}}^K \left( \mathbf{v}^E \right) = \exp_{\mathbf{o}}^K \left( [0, \mathbf{v}^E] \right) = \left( \underbrace{\cosh \left( \frac{\|\mathbf{v}^E\|_2}{\sqrt{K}} \right)}_{\text{time-like dimension } x_t}, \underbrace{\sqrt{K} \sinh \left( \frac{\|\mathbf{v}^E\|_2}{\sqrt{K}} \right) \frac{\mathbf{v}^E}{\|\mathbf{v}^E\|_2}}_{\text{space-like dimensions } \mathbf{x}_s} \right). \quad (1)$$

**Fully Lorentz Neural Networks**  Fully Lorentz networks (Chen et al., 2021) are proved to be ideal for PFL due to their reduced need for space projections, enhancing computational efficiency. These networks also incorporate Lorentz transformations (boosts and rotations), improving data heterogeneity handling and parameter interpretability (Appendix A.3).

Given an input vector $\mathbf{x} \in \mathcal{L}_K^n$, and a linear layer matrix $\hat{\mathbf{M}} \in \mathbb{R}^{(m+1) \times (n+1)}$ to optimize, $\forall \mathbf{x} \in \mathcal{L}_K^n, \hat{\mathbf{M}}\mathbf{x} \in \mathcal{L}_K^m$. Let $\hat{\mathbf{M}} = \begin{bmatrix} \mathbf{v}^T \\ \mathbf{W} \end{bmatrix}, \mathbf{v} \in \mathbb{R}^{(n+1)}, \mathbf{W} \in \mathbb{R}^{m \times (n+1)}$. The fully Lorentz linear layer can be denoted as LT in a general form as follows:

---

[3]For clarity, all Lorentz space embeddings are denoted by $\cdot^H$. Specifically, if the curvature of the space is known as $K$, it is denoted by $\cdot^K$. In contrast, Euclidean space embeddings are denoted by $\cdot^E$.

$$\text{LT}(\mathbf{x}; f; \mathbf{W}) := \left( \sqrt{\|f(\mathbf{W}\mathbf{x}, \mathbf{v})\|^2 + K}, f(\mathbf{W}\mathbf{x}, \mathbf{v}) \right)^T. \tag{2}$$

It involves a function $f$ that operates on vectors $\mathbf{v} \in \mathbb{R}^{n+1}$ and $\mathbf{W} \in \mathbb{R}^{m \times (n+1)}$. Depending on the type of function, it can perform different operations. For instance, for dropout, the operation function is $f(\mathbf{W}\mathbf{x}, \mathbf{v}) = \mathbf{W}$ dropout $(\mathbf{x})$. For normalization with learned scale, $f(\mathbf{W}\mathbf{x}, \mathbf{v}) = \frac{\sigma(\mathbf{v}^T \mathbf{x})}{\|\mathbf{W}\mathbf{x}\|} \mathbf{W}\mathbf{x}$.

## 4 MOTIVATION AND INSIGHTS

This paper focuses on graph data for its clear distribution and simpler models, facilitating the validation of our approach using Lorentz neural networks to address heterogeneity in personalized federated learning. Our method is also applicable to other datasets and tasks.

### PROBLEM STATEMENT

Given clients $\mathcal{C} = 1, 2, \ldots, C$, each with a dataset $\mathcal{D}_c = (\mathbf{x}_i^c, y_i^c)_{i=1}^{N_c}$ and distribution $p_c(\mathbf{x}, y)$, Personalized Federated Learning (PFL) encounters distributional heterogeneity if $p_i(\mathbf{x}, y) \neq p_j(\mathbf{x}, y)$ for any clients $i \neq j$. This heterogeneity can degrade performance. In PFL, the goal is to optimize personalized models $f_c(\cdot; \boldsymbol{\theta}_c, \boldsymbol{\theta}_s)$ for each client using specific and shared parameters $\boldsymbol{\theta}_c, \boldsymbol{\theta}_s$.

$$\min_{\boldsymbol{\theta}_c|_{c=1}^C, \boldsymbol{\theta}_s} \sum_{c=1}^C \mathbb{E}_{(\mathbf{x}, y) \sim p_c(\mathbf{x}, y)} [\mathcal{L}_c(f(\mathbf{x}; \boldsymbol{\theta}_c, \boldsymbol{\theta}_s), y)] + \lambda \Omega(\boldsymbol{\theta}_c|_{c=1}^C, \boldsymbol{\theta}_s) \tag{3}$$

This function merges local loss $\mathcal{L}_c$ with regularization $\Omega$, balanced by hyperparameter $\lambda$.

---

**Our goals** are

(1) to *effectively* represent the inherent properties of each local client data;

(2) to *succinctly* reflect heterogeneity among client data and facilitate the communication of shared information without requiring additional computations.

---

INSIGHTS: INTRODUCE A HIGHER DIMENSION (*time axes*) TO *"Flatland"*.

---

*In "Flatland", a two-dimensional flat plane, the same shapes may represent the projections of various three-dimensional objects. For instance, a circle could be the projection of either a cylinder or a sphere from a higher dimension.*

---

In the above case, *"Flatland"* captures the common feature of a cylinder or a sphere, while a higher dimension (the third dimension) highlights the differences between the objects. Analogous to our setting, informally speaking, by introducing an additional *time-like* dimension, we can imagine each client's data residing in a unique Lorentz space (a curved world in a higher-dimensional space), where the curvature reflects the distinct distributions (objects). *"Flatland"*, $\mathbb{R}^d$ (flat), serves as a metaphor for a platform where common information (circle) is exchanged and integrated.

MOTIVATION: WHY LORENTZ SPACE?

(1) Prevalent Non-Euclidean properties of real-world data. Forman-Ricci curvature $\overline{\text{Ric}}$ measures deviations from flat (Euclidean) geometry in data structures (Sandhu et al., 2016; Forman, 2003). A more negative $\overline{\text{Ric}}$ indicates a structure more suited for hyperbolic space representation (Sun et al., 2024). Figure 2 shows varying $\overline{\text{Ric}}$ values across 10 clients from the CiteSeer dataset, highlighting the common non-Euclidean nature of real-world data. Thus, employing Lorentz space with client-specific curvature can better capture intrinsic data structures, supporting our goal (1).

(2) Strong correlation between heterogeneity and curvature. Figure 1(a) shows that distribution curves exhibit long-tailed characteristic with varying skewness, supporting the findings from previous

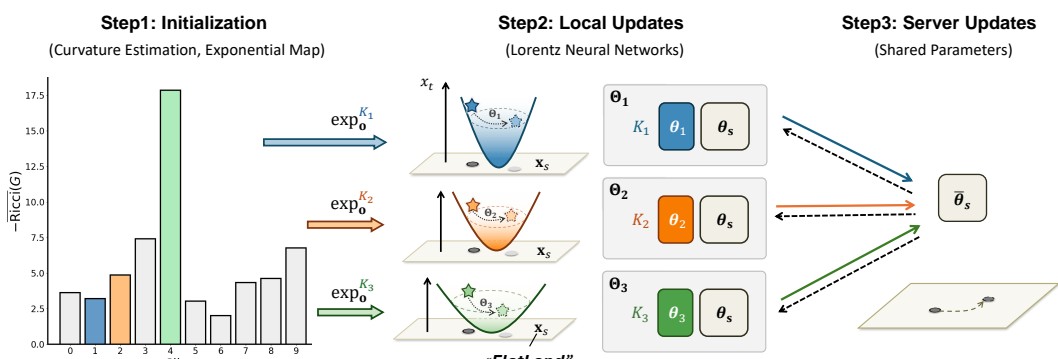

Figure 2: The FlatLand framework.

studies (Xie et al., 2021). In particular, Client 1's distribution is steeper and less Euclidean, suggesting a need for embedding in a Lorentz space with a larger curvature (a smaller $K$), depicted in Figure 1(b). This space accommodates more tail nodes (black stars) than Clients 2 and 3, requiring a "roomier" embedding environment to ensure separability and enhance performance. A larger curvature facilitates this by allowing embeddings to occupy a "higher" position (larger $x_t$) in the space, where the volume expands exponentially.

The observations align with our **goal** (2) because heterogeneous properties like "*how significant is the imbalance between tail nodes and head nodes?*" can be naturally distinguished through their corresponding Lorentz space with different curvature (differed by the *time-like* axes $x_t$). Meanwhile, when the star nodes are mapped back to the Euclidean space, the common information, e.g., *"the star is the tail node in their client"*, is preserved in *space-like* dimensions $\mathbf{x}_s$ as the same node $\mathbf{v}$.

## 5   THE FlatLand FRAMEWORK

We propose a personalized federated learning framework, FlatLand, using tailored Lorentz spaces for each client. The main steps are outlined in Figure 2 and Algorithm 2.

- **S1 Initialization.** At the initial communication round $r = 0$, the parameters that need to be initialized can be divided into three parts:

    (1) Curvature parameters of $C$ clients $\{K_1, K_2, ...K_C\}$ ;               (**Section 5.1**)
    (2) Personalized parameters of $C$ clients $\{\boldsymbol{\theta}_1, \boldsymbol{\theta}_2, ..., \boldsymbol{\theta}_C\}$;               (**Section 5.2**)
    (3) Shared parameters $\overline{\boldsymbol{\theta}}_s$ of central server.

    All the parameters of client $i$ at round 0 can be written as $\Theta_i^{(0)} = \left(K_i; \boldsymbol{\theta}_i^{(0)}; \overline{\boldsymbol{\theta}}_s^{(0)}\right)$ and server parameters as $\overline{\boldsymbol{\theta}}_s^{(0)}$.

- **S2 Local updates.** Given learning rate $\eta$, for round $r$, each local client model performs training on the data $\mathcal{D}_i$ to minimize the task loss $\mathcal{L}(\mathcal{D}_i; \Theta_i^{(r)})$ and then updating the parameters as $\Theta_i^{(r+1)} \leftarrow \Theta_i^{(r)} - \eta\nabla\mathcal{L}.$               (**Section 5.3**)

- **S3 Server updates.** After local training, only shared parameters $\boldsymbol{\theta}_{s_c}^{(r+1)}$ are updated to the server for each client $c$. These are then aggregated using FedAvg: $\overline{\boldsymbol{\theta}}_s^{(r+1)} \leftarrow \frac{N_c}{N} \sum_{c=1}^{C} \boldsymbol{\theta}_{s_c}^{(r+1)}$, , where $N = \sum_c N_c$. The aggregated parameters are subsequently distributed to clients for the next round.

### 5.1   CURVATURE ESTIMATION

To embed the dataset $\mathcal{D}_c$ of client $c \in \mathcal{C}$ into its tailored Lorentz space $\mathcal{L}_{K_c}^d$, a suitable curvature $K_c$ should be first explored.

There are many comprehensive ways can assist in estimating the suitable curvature for various types of data (Gao et al., 2021). Here, given a weighted graph $G_c = (V, E, w)$ in client $c$, we adopt Forman-Ricci curvature (Appendix A.2) and the overall curvature of the graph can be calculated as follows $\overline{\text{Ric}}(G) = \frac{1}{|E|} \sum_{(x,y) \in E} \text{Ric}(x, y)$, where $V$ represents graph nodes and $|E|$ the number of edges, specifically, $(x, y)$ means the edge between node $x$ to node $y$. Additionally, the curvature can be a learnable parameter or calculated using a simple Multi-Layer Perceptron (MLP) neural network. Here, we initialize $K_c$ with $\overline{\text{Ric}}(G_c)$ as learnable.

## 5.2 Parameter Decoupling Strategy

This section details the fully Lorentz model's parameters (excluding $K$), divided into shared $\boldsymbol{\theta}_s$ for *space-like* dimensions and personalized $\boldsymbol{\theta}_c$ for *time-like* dimension. The model has layers of fully Lorentz neural networks that transform data within Lorentz space (Section 3).

First, without loss of generality, we decouple the function of Lorentz linear layer in Equation (2) without the functions $f$ of activation, dropout, bias, and so on.

Given input $\mathbf{x}^{(l)} = \begin{bmatrix} x_t^{(l)} \\ \mathbf{x}_s^{(l)} \end{bmatrix} \in \mathcal{L}_K^n, x_t^{(l)} \in \mathbb{R}, \mathbf{x}_s^{(l)} \in \mathbb{R}^n$ in layer $l$. We rewrite the learnable matrix $\hat{\mathbf{M}}^{(l)}$ in Section 3 as $\begin{bmatrix} v^{(l)} & \mathbf{v}^{T(l)} \\ m^{(l)} & \mathbf{M}^{(l)} \end{bmatrix} \in \mathbb{R}^{(m+1) \times (n+1)}, v^{(l)} \in \mathbb{R}, \mathbf{v}^{(l)} \in \mathbb{R}^n, m^{(l)} \in \mathbb{R}^m, \mathbf{M}^{(l)} \in \mathbb{R}^{m \times n}$, the output $\mathbf{x}^{(l+1)}$ of the Lorentz linear layer could be reformulated as

$$\mathbf{x}^{(l+1)} = \text{LT}(\mathbf{x}^{(l)}; \hat{\mathbf{M}}^{(l)}) = \left( \underbrace{\sqrt{\|mx_t + \mathbf{M}\mathbf{x}_s\|^2 + K}}_{\text{time-like dimension } x_t^{(l+1)}}, \quad \underbrace{mx_t + \mathbf{M}\mathbf{x}_s}_{\text{space-like dimensions } \mathbf{x}_s^{(l+1)}} \right)^T. \quad (4)$$

Then, we decouple the parameters as follows under the deviation from Appendix B.3:

> Suppose the model $\mathcal{M}$ consists of $L$ layers of neural networks,
> - The personalized parameter set $\boldsymbol{\theta}_c$ for all layers is formulated as
> $$\boldsymbol{\theta}_c = \bigcup_{l=1}^{L} \{v^{(l)}, \mathbf{v}^{T(l)}, m^{(l)}\};$$
> - The shared parameter set $\boldsymbol{\theta}_s$ across all layers is formulated as
> $$\boldsymbol{\theta}_s = \bigcup_{l=1}^{L} \{M^{(l)}\};$$
> where $\bigcup_{l=1}^{L}$ indicates the union of parameter sets from each layer $l$ from 1 to $L$.

## 5.3 Local Training Procedure

Obtained the curvature $K_c^{(r)}$ at round $r$, we directly project the client input $\mathbf{x}_i^E \in \mathcal{D}_c$ into its corresponding Lorentz space via the exponential map $\mathbf{x}^{K_c} = \exp_{\mathbf{o}}^{K_c}(\mathbf{x}^E)$, as shown in Equation (1). Note that to simplify the notation, all vectors $\mathbf{x}$, if not superscripted, are assumed to represent being in the Lorentz space.

Afterwards, the training data are fed into the Lorentz model $\mathcal{M}$, the output is $f((\mathbf{x}^{K_c}; \boldsymbol{\theta}_c, \boldsymbol{\theta}_s), y)$. In the graph model, in addition to the Lorentz linear layer, there is also an aggregation operation (Zhang et al., 2021c), which does not involve any parameters, so it has no impact on our results.

At client $c$, the objective function is

$$\min_{\boldsymbol{\theta}_c|_{c=1}^{C},\boldsymbol{\theta}_s} \mathcal{L}_c(f(\mathbf{x}^{K_c}; \boldsymbol{\theta}_c, \boldsymbol{\theta}_s), y) + \lambda \|\boldsymbol{\theta}_{s_c} - \overline{\boldsymbol{\theta}}_s\|_2^2, \tag{5}$$

where $\lambda$ is a hyperparameter, $\|\boldsymbol{\theta}_{s_c} - \overline{\boldsymbol{\theta}}_s\|_2^2$ is the regularize term that prevent locally updated model $\boldsymbol{\theta}_{s_c}$ deviates too far from the server shared parameters $\overline{\boldsymbol{\theta}}_s$.

## 6 ANALYSIS

In this section, we provide further analysis to demonstrate the effectiveness and interpretability of our method as described in Section 5.2. Specifically, we first verify the **correctness** that federated learning does not cause the data in each client to deviate from its original space during the process of parameter communication (server updates). Furthermore, we expound on the rationale behind our proposed method from the perspectives of debiasing and Lorentz transformation.

**Proposition 1.** $\forall \mathbf{x} \in \mathcal{L}_K^n, \forall \mathbf{M} \in \mathbb{R}^{(m+1)\times(n+1)}$, we have $\mathrm{LT}(\mathbf{x}; \mathbf{M}) \in \mathcal{L}_K^m$.

*Proof.* $\forall \mathbf{x} \in \mathcal{L}_K^n$, we have $\langle \mathrm{LT}(\mathbf{x}; \mathbf{M}), \mathrm{LT}(\mathbf{x}; \mathbf{M}) \rangle_{\mathcal{L}} = -K$. Therefore, $\mathrm{LT}(\mathbf{x}; \mathbf{M}) \in \mathcal{L}_K^m$. $\qquad \square$

**Corollary 1.** *Let* $\hat{\mathbf{M}} = \begin{bmatrix} v & \mathbf{v}^T \\ m & \mathbf{M} \end{bmatrix}$, *where* $\hat{\mathbf{M}} \in \mathbb{R}^{(m+1)\times(n+1)}$ *and* $\Phi\left(\hat{\mathbf{M}}, \mathbf{N}\right) = \begin{bmatrix} v & \mathbf{v}^T \\ m & \mathbf{N} \end{bmatrix}. \forall \mathbf{x} \in \mathcal{L}_K^n, \forall \hat{\mathbf{M}} \in \mathbb{R}^{(m+1)\times(n+1)}, \forall \mathbf{N} \in \mathbb{R}^{n\times n}$, *we have* $\mathrm{LT}\left(\mathbf{x}; \Phi\left(\hat{\mathbf{M}}, \mathbf{N}\right)\right) \in \mathcal{L}_K^m$.

This corollary (refer to the proof in the Appendix B.4) implies that even after the aggregation of shared parameters in the server, the transformation of any client vector $\mathbf{x} \in \mathcal{L}_K^n$ by this updated matrix will still yield results in the Lorentz space $\mathcal{L}_K^m$ with the same curvature, indicating that the client's representation remains unaffected.

### PERSPECTIVES ON DEBIASING

**Remark 1** (Feature Debiasing). *During the local and server updates in* FlatLand, *the debiasing process is inherently integrated via the gradient of shared parameters* $\mathbf{M}$.

According to the derivations in Appendix B.3, it can be observed that the gradient of the shared parameters $\mathbf{M}$ is highly correlated with $\mathbf{x}_s$, where $\mathbf{x}_s$ is derived from the raw input $\mathbf{x}^E$ using the exponential map in Equation (1). Therefore, given the same input $\mathbf{x}^E$ for different clients tailored to different Lorentz manifolds, the gradient of $\mathbf{M}$ for client $c$ is inherently weighted by $\sqrt{K_c} \sinh\left(\frac{\|\mathbf{x}^E\|_2}{\sqrt{K_c}}\right) \frac{1}{\|\mathbf{x}^E\|_2}$, where $K_c$ can be intuitively interpreted as the parameter that reflects the overall distribution of the dataset specific to client $c$, which differs from other clients. This can play a role in debiasing during the parameter aggregation process compared to Euclidean methods.

### PERSPECTIVES ON LORENTZ TRANSFORMATIONS

Lorentz Boosts and Lorentz Rotations (Appendix A.3) are interpreted as being covered by $\mathrm{LT}\left(\mathbf{x}; \hat{\mathbf{M}}\right)$ when the dimension is unchanged (Chen et al., 2021). We can easily prove that the Lorentz transformations are still covered by $\mathrm{LT}\left(\cdot; \Phi\left(\hat{\mathbf{M}}, \mathbf{N}\right)\right)$, where $\hat{\mathbf{M}} \in \mathbb{R}^{(n+1)\times(n+1)}, \mathbf{N} \in \mathbb{R}^{n\times n}$.

For any data point $\mathbf{x} \in \mathcal{D}_c$, transformations $\mathrm{LT}\left(\mathbf{x}; \hat{\mathbf{M}}\right)$ and $\mathrm{LT}\left(\mathbf{x}; \Phi\left(\hat{\mathbf{M}}, \mathbf{N}\right)\right)$ map $\mathbf{x}$ to a new spacetime position, maintaining the spacetime interval invariant (Corollary 1), thus preserving the physical and geometric relationships within the same client, in line with special relativity. However, clients with varying spacetime curvatures maintain **distinct spacetime intervals**, reflecting differing underlying data distributions.

Moreover, according to the definition of Lorentz Rotation in Equation (9), the server updates only the $\mathbf{M}$, leaving the time-like dimension unchanged. This operation is a relaxation of the Lorentz rotation, consistent with our "Flatland" assumption that aggregates only spatial dimension information.

Table 2: Comparison of node classification performance across real-world datasets with varying numbers of clients. The results, presented as mean and standard deviation, are based on five separate trials. Performances that are statistically significant ($p < 0.05$) are highlighted in bold.

| | Cora | | CiteSeer | | ogbn-arxiv | | Photo | |
|---|---|---|---|---|---|---|---|---|
| # clients | 10 | 20 | 10 | 20 | 10 | 20 | 10 | 20 |
| Local ($E$) | $79.94 \pm 0.24$ | $80.30 \pm 0.25$ | $67.82 \pm 0.13$ | $65.98 \pm 0.17$ | $64.92 \pm 0.09$ | $65.06 \pm 0.05$ | $91.80 \pm 0.02$ | $90.47 \pm 0.15$ |
| Local ($L$) | $78.35 \pm 0.05$ | $80.46 \pm 0.18$ | $72.30 \pm 0.04$ | $69.52 \pm 0.25$ | $65.85 \pm 0.09$ | $66.75 \pm 0.05$ | $91.76 \pm 0.10$ | $90.12 \pm 0.20$ |
| FedAvg | $69.19 \pm 0.67$ | $69.50 \pm 3.58$ | $63.61 \pm 3.59$ | $64.68 \pm 1.83$ | $64.44 \pm 0.10$ | $63.24 \pm 0.13$ | $83.15 \pm 3.71$ | $81.35 \pm 1.04$ |
| FedPer | $79.35 \pm 0.04$ | $78.01 \pm 0.32$ | $70.53 \pm 0.28$ | $66.64 \pm 0.27$ | $64.99 \pm 0.18$ | $64.66 \pm 0.11$ | $91.76 \pm 0.23$ | $90.59 \pm 0.06$ |
| FedProx | $60.18 \pm 7.04$ | $48.22 \pm 6.81$ | $63.33 \pm 3.25$ | $64.85 \pm 1.35$ | $64.37 \pm 0.18$ | $63.03 \pm 0.04$ | $80.92 \pm 4.64$ | $82.32 \pm 0.29$ |
| FedGNN | $70.12 \pm 0.99$ | $70.10 \pm 3.52$ | $55.52 \pm 3.17$ | $52.23 \pm 6.00$ | $64.21 \pm 0.32$ | $63.80 \pm 0.05$ | $87.12 \pm 2.01$ | $81.00 \pm 4.48$ |
| FedSage+ | $69.05 \pm 1.59$ | $57.97 \pm 12.6$ | $65.63 \pm 3.10$ | $65.46 \pm 0.74$ | $64.52 \pm 0.14$ | $63.31 \pm 0.20$ | $76.81 \pm 8.24$ | $80.58 \pm 1.15$ |
| GCFL | $78.66 \pm 0.27$ | $79.21 \pm 0.70$ | $69.01 \pm 0.12$ | $66.33 \pm 0.05$ | $65.09 \pm 0.08$ | $65.08 \pm 0.04$ | $92.06 \pm 0.25$ | $90.79 \pm 0.17$ |
| FedHGCN | $72.09 \pm 0.16$ | $74.67 \pm 1.50$ | $66.98 \pm 0.56$ | $64.28 \pm 0.62$ | OOM | OOM | $79.26 \pm 0.56$ | $79.57 \pm 0.10$ |
| **FlatLand (Ours)** | $\mathbf{80.46 \pm 0.28}$ | $\mathbf{82.49 \pm 0.25}$ | $\mathbf{73.90 \pm 0.23}$ | $\mathbf{72.24 \pm 0.24}$ | $\mathbf{67.52 \pm 0.16}$ | $\mathbf{67.64 \pm 0.04}$ | $\mathbf{92.49 \pm 0.19}$ | $\mathbf{91.06 \pm 0.15}$ |

# 7 EXPERIMENTS

In this section, we validate the effectiveness of FlatLand by conducting experiments for *node classification* and *graph classification* on a series of benchmark datasets. The experiments are designed to address the following research questions. **RQ1.** Can FlatLand outperform personalized and hyperbolic FL baselines? **RQ2.** Can FlatLand still perform well in low-dimensional settings? **RQ3.** Are the proposed novel components really beneficial?

## 7.1 EXPERIMENTAL SETUP

**Datasets and Baselines** The details about datasets are listed in Appendix C.1. Implementation details are shown in Appendix C.2. More detailed information can be found in our anonymous repository. To assess FlatLand and demonstrate its superiority, we compare it with the following baselines: (1) Local: clients train their models locally without any communication, Local ($E$) refers to self-training in the Euclidean model, while Local ($L$) refers to training in the Lorentz model.; (2) FedAvg (McMahan et al., 2017) and (3) FedProx (Li et al., 2020a): the most popular FL baselines; (4) FedPer (Arivazhagan et al., 2019): a PFL baseline with personalized model layers; (5) FedGNN (Wu et al., 2021) and (6) FedSage (Zhang et al., 2021a): two FGL baselines; (7) GCFL (Xie et al., 2021): a PFGL baseline with client clustering and cluster-wise model aggregation; (8) FedHGCN (Du et al., 2024): a hyperbolic FGL baseline that fails considering the heterogeneity among clients.

## 7.2 MAIN EXPERIMENTAL RESULTS (RQ1)

**Node Classification** We tackle node classification on *highly heterogeneous datasets*, with non-overlapping node partitions for each client, which most previous work fail to address. This challenge highlights our method's ability to handle heterogeneity that previous approaches could not address. Table 2 shows that our proposed FlatLand outperforms all baselines with statistical significance ($p < 0.05$). (1) Local ($L$) often surpasses Local ($E$), suggesting that hyperbolic space can better represent most datasets, though the gap is sometimes marginal. (2) Euclidean FL methods like FedAvg, FedProx, FedGNN, and FedSage+ significantly underperform self-training. GCFL is generally the best among Euclidean methods, but cannot consistently beat Local ($E$). FedPer sometimes exceeds Local ($E$) with small

Table 1: Performance on graph classification tasks. The results, presented as mean and standard deviation, are based on five separate trials. Performances that are statistically significant ($p < 0.05$) are highlighted in bold.

| | CHEM (1) | BIO-CHEM-SN (3) |
|---|---|---|
| # datasets | 7 | 13 |
| Local ($E$) | $75.54 \pm 1.73$ | $67.17 \pm 1.76$ |
| Local ($L$) | $75.72 \pm 2.41$ | $65.31 \pm 2.13$ |
| FedAvg | $75.88 \pm 2.17$ | $66.91 \pm 1.94$ |
| FedProx | $76.05 \pm 1.92$ | $66.34 \pm 2.26$ |
| FedPer | $75.81 \pm 2.17$ | $66.27 \pm 2.09$ |
| GCFL | $76.49 \pm 1.23$ | $67.21 \pm 2.39$ |
| FedHGCN | $75.06 \pm 1.81$ | OOM |
| **FlatLand (Ours)** | $\mathbf{76.55 \pm 2.28}$ | $\mathbf{67.31 \pm 2.58}$ |

gains, highlighting challenges with heterogeneous data. (3) FedHGCN, despite operating in hyperbolic space, underperforms on heterogeneous datasets by not accounting for data heterogeneity, akin to FedAvg vs Local ($E$) in Euclidean space. Besides, due to the quadratic time and space

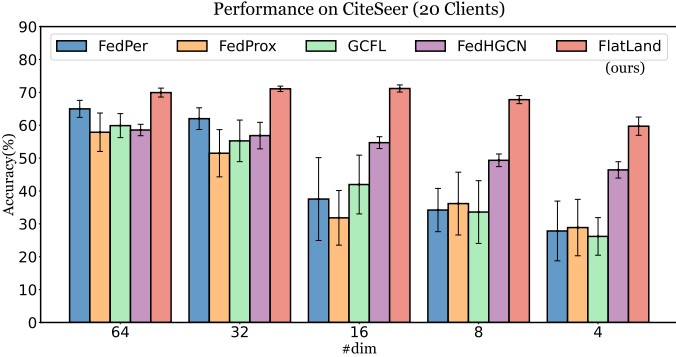
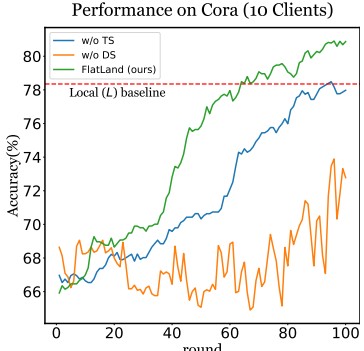

Figure 3: Performance of CiteSeer (20 clients) with varying dimensions for node classification scenario.

Figure 4: Ablation study of FlatLand on the Cora dataset.

complexity of FedHGCN's node selection module. Therefore, it can easily encounter out-of-memory (OOM) issues with large datasets, like ogbn-arxiv. In conclusion, experiments show that FlatLand can mitigate the heterogeneity, and with larger gains on highly heterogeneous datasets like CiteSeer.

**Graph Classification**  Table 3 shows the results of the graph classification task, which is conducted with multiple datasets from one or more domains owned by different clients in each task/setting. In the single-dataset CHEM setting, Local ($L$) outperforms Local ($E$) due to inherent hyperbolic characteristics better captured by hyperbolic geometry. However, in multiple-dataset settings like BIO-CHEM-SN, Local ($L$) fails to surpass Local ($E$), potentially because not all datasets exhibit prominent hyperbolic features. With our proposed federated graph learning approach, FlatLand can significantly enhance the performance of the Lorentzian model, outperforming the Euclidean baselines, and demonstrating the effectiveness of our proposed method.

**Convergence Curves**  The convergence curves for node classification tasks are shown in Figure 7 in Appendix C.5. As the figures demonstrate, our proposed method has great convergence speed, highlighting the superiority of our proposed approach.

## 7.3 VARYING EMBEDDING DIMENSIONS (RQ2)

Lower embedding and hidden dimensions reduce the parameter transmission cost in federated learning, as fewer parameters are communicated between the server and clients during training. Considering the representational power of hyperbolic spaces in lower dimensions (Chami et al., 2019), we reduced the embedding dimension from 64 to 4 to evaluate FlatLand's ability to mitigate data heterogeneity using compact representations. Figure 3 shows the results on CiteSeer (20 clients), with similar trends observed across datasets. Dimensionality reduction from 64 to 4 had a relatively small impact on the hyperbolic methods (FlatLand and FedHGCN) compared to their Euclidean counterparts. Notably, while FedHGCN underperformed Euclidean methods at higher dimensions, it outperformed them when the dimension was reduced to 16. FlatLand consistently outperformed all other methods in different embedding dimensions, and its performance advantage over the baselines became increasingly significant as the dimensionality was reduced.

## 7.4 ABLATION STUDY (RQ3)

To analyze the contribution of each component, we conduct ablation studies. Figure 4. Through ablation studies, we analyze the contribution of each component to the model's performance.

**The benefits of adaptive curvature**  The "w/o TS" (without tailed space) refers to setting a constant curvature of 1 for all clients instead of employing tailored curvature settings. It indicates that using a fixed hyperbolic space with constant curvature yields inferior performance compared to utilizing tailored curvatures. Furthermore, the results obtained with tailored curvatures closely approximate those of the local ($L$) setting, demonstrating the inherent effectiveness of the hyperbolic space itself.

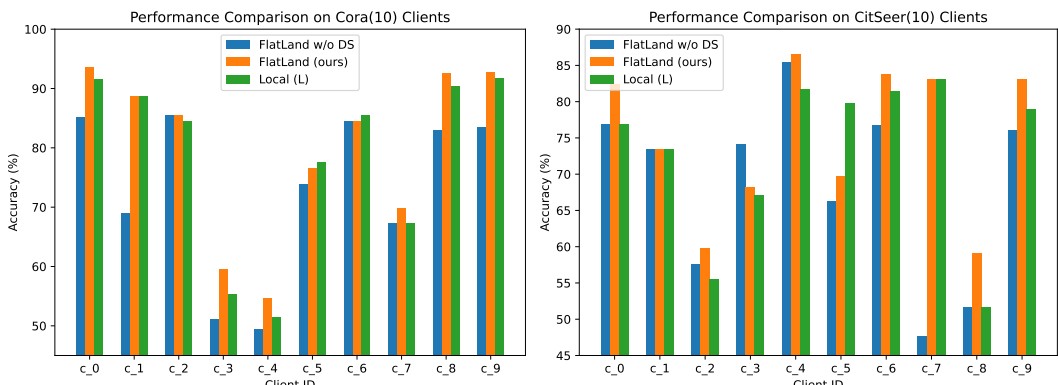

Figure 5: Performance comparison of FlatLand on Cora and CitSeer across local 10 clients.

**The benefits of time-like parameters decoupling.** The "w/o DS" refers to no **parameter decoupling strategy**, which exhibits significant fluctuations across rounds because the aggregation process incorporates heterogeneous information, adversely affecting the results. This highlights the effectiveness of our proposed decoupling strategy and validates that the time-like dimension can effectively capture heterogeneous information. Moreover, we analyze the benefits of DS for each client's performance. As shown in Figure 5, with client IDs on the x-axis, Flatland outperforms the local method for the vast majority clients, notably improving performance for clients with inherently poorer results, like $c\_8$ in the CiteSeer dataset. This underscores *the necessity of federated settings for hyperbolic models*. Without our proposed DS, performance deteriorates significantly (e.g., $c\_7$ in CiteSeer), further *validating our hypothesis that the time-like parameter encapsulates crucial heterogeneity information.*

**The necessity of Lorentz space** We conducted experiments to further evaluate the necessity of using Lorentz space. Table 3 presents the results of an ablation study on the Lorentz transformation. FlatLand ($E$) represents our proposed method with parameter decoupling strategy implemented using an Euclidean backbone. Without Lorentz geometry, FlatLand ($E$) underperforms because the time-like parameter loses its geometric meaning. It even falls short of FedPer in most cases, which uses the classifier layer for personalization. These results validate our hypothesis and underscore the importance of hyperbolic representation for our proposed decoupling strategy in our method.

## 8 CONCLUSION AND LIMITATIONS

Table 3: Ablation study results about the necessity of using Lorentz space to do parameter decoupling.

|  | Cora (10) | Cora (20) | CiteSeer (10) | CiteSeer (20) |
|---|---|---|---|---|
| # datasets | 10 | 20 | 10 | 20 |
| FedAvg | $69.19 \pm 0.67$ | $69.50 \pm 3.58$ | $63.61 \pm 3.59$ | $64.68 \pm 1.83$ |
| FedPer | $\underline{79.35} \pm 0.04$ | $\underline{78.01} \pm 0.32$ | $70.53 \pm 0.28$ | $\underline{66.64} \pm 0.27$ |
| FlatLand ($E$) | $78.53 \pm 0.73$ | $76.23 \pm 0.43$ | $\underline{70.68} \pm 0.52$ | $66.29 \pm 0.35$ |
| FlatLand (ours) | $\mathbf{80.46} \pm 0.28$ | $\mathbf{82.49} \pm 0.25$ | $\mathbf{73.90} \pm 0.23$ | $\mathbf{72.24} \pm 0.24$ |

**Conclusions** In this paper, we introduce FlatLand, an exploratory personalized federated learning approach leveraging hyperbolic geometry to succinctly capture heterogeneity across clients' data distributions embedded in tailored Lorentz spaces. We propose a novel parameter decoupling strategy, which enables server-side aggregation of common information while mitigating heterogeneity interference, without client similarity estimation. This is a previously unexplored approach not only in FL but also in hyperbolic geometry. As the first work incorporating hyperbolic geometry into PFL, FlatLand demonstrates superior performance over Euclidean counterparts, especially in low dimensions, showcasing strong potential as an effective solution to the heterogeneity challenge.

**Future work** While evaluated on graph data, FlatLand is not limited to graphs and can be extended to other data types. Note that hyperbolic space is not universally optimal for all data distributions — some exhibit positive curvature — highlighting the need to model complex data structures in mixed-curvature spaces. Moreover, more complex Lorentz neural networks can be explored for federated learning of sophisticated models beyond the simple encoder used currently. Therefore, our next step is to extend and evaluate FlatLand to more complex backbones and tasks.

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

# Contents

# APPENDIX / SUPPLEMENTAL MATERIAL

## A    PRELIMINARIES

### A.1    LORENTZ MANIFOLD: FORMAL DEFINITIONS

Hyperbolic space is non-Euclidean geometry with a constant negative curvature. The curvature of hyperbolic space is a measure of how the geometry of the space deviates from the flatness of Euclidean space. The Lorentz manifold, also known as the hyperboloid model, is one of the most commonly used mathematical representations of hyperbolic space. Its greater stability for numerical optimization makes it a popular choice for hyperbolic geometry methods Nickel & Kiela (2018).

**Definition 1** (Lorentz Manifold). *A $d$-dimensional Lorentz manifold $\mathcal{L}_K^d$ with a negative curvature of $-1/K(K > 0)$ can be defined as the Riemannian manifold $\left(\mathbb{H}_K^d, g_\ell\right)$, where $g_\ell = \mathrm{diag}([-K, 1, \ldots, 1])$ and $\mathbb{H}_K^d = \left\{\mathbf{x} \in \mathbb{R}^{d+1} : \langle \mathbf{x}, \mathbf{x}\rangle_{\mathcal{L}} = -K, x_0 > 0\right\}.$*

**Definition 2** (Lorentzian Inner Product). *The inner product $\langle \mathbf{x}, \mathbf{y}\rangle_{\mathcal{L}}$ for $\mathbf{x}, \mathbf{y} \in \mathbb{R}^{d+1}$ can be defined as let $\langle \mathbf{x}, \mathbf{y}\rangle_{\mathcal{L}} = -x_0 y_0 + \sum_{i=1}^d x_d y_d.$*

Based on the constraint $\langle \mathbf{x}, \mathbf{x}\rangle_{\mathcal{L}} = -K$, it holds for any point $\mathbf{x} = (x_0, \mathbf{x}') \in \mathbb{R}^{d+1}$ that $\mathbf{x} \in \mathcal{L}_K^d \Leftrightarrow x_0 = \sqrt{\|\mathbf{x}'\| + K}$. The larger the value of $K$, the greater the extent to which the hyperbolic surface deviates from the Euclidean plane, as it is influenced by the larger value of $x_0$.

Next, the corresponding Lorentzian distance function for two points $\mathbf{x}, \mathbf{y} \in \mathcal{L}_K^d$ is provided as

$$d_{\mathcal{L}}^K(\mathbf{x}, \mathbf{y}) = \sqrt{K}\mathrm{arcosh}(-\langle \mathbf{x}, \mathbf{y}\rangle_{\mathcal{L}}/K). \tag{6}$$

**Definition 3** (Tangent Space). *For a point $\mathbf{x} \in \mathcal{L}_K^d$, the tangent space $\mathcal{T}_{\mathbf{x}}\mathcal{L}_K^d$ consists of all vectors orthogonal to $\mathbf{x}$, where orthogonality is defined with respect to the Lorentzian inner product( Definition 2). Hence, $\mathcal{T}_{\mathbf{x}}\mathcal{L}_K^d = \left\{\mathbf{v} : \langle \mathbf{x}, \mathbf{v}\rangle_{\mathcal{L}} = 0\right\}.$*

**Definition 4** (Exponential and Logarithmic Maps). *Let $\mathbf{v} \in \mathcal{T}_x\mathcal{L}_K^d$. The exponential map $\exp_{\mathbf{x}}^K : \mathcal{T}_{\mathbf{x}}\mathcal{L}_K^d \rightarrow \mathcal{L}_K^d$ and logarithmic map $\log_{\mathbf{x}}^K : \mathcal{L}_K^d \rightarrow \mathcal{T}_{\mathbf{x}}\mathcal{L}_K^d$ are defined as*

$$\exp_{\mathbf{x}}^K(\mathbf{v}) = \cosh\left(\frac{\|\mathbf{v}\|_{\mathcal{L}}}{\sqrt{K}}\right)\mathbf{x} + \sqrt{K}\sinh\left(\frac{\|\mathbf{v}\|_{\mathcal{L}}}{\sqrt{K}}\right)\frac{\mathbf{v}}{\|\mathbf{v}\|_{\mathcal{L}}},$$

$$\log_{\mathbf{x}}^K(\mathbf{y}) = d_{\mathcal{L}}^K(\mathbf{x}, \mathbf{y})\frac{\mathbf{y} + \frac{1}{K}\langle \mathbf{x}, \mathbf{y}\rangle_{\mathcal{L}}\mathbf{x}}{\left\|\mathbf{y} + \frac{1}{K}\langle \mathbf{x}, \mathbf{y}\rangle_{\mathcal{L}}\mathbf{x}\right\|_{\mathcal{L}}},$$

*where $\|\mathbf{v}\|_{\mathcal{L}} = \sqrt{\langle \mathbf{v}, \mathbf{v}\rangle_{\mathcal{L}}}$ denotes the norm of $\mathbf{v}$ in $\mathcal{T}_{\mathbf{x}}\mathcal{L}_K^d$.*

Particularly, for the sake of calculation, the origin of Lorentz manifold $\mathbf{o} = (\sqrt{K}, 0, 0, ..., 0) \in \mathcal{L}_K^d$ is chosen as the reference point for the exponential and logarithmic maps, which can be simplified as

$$\exp_{\mathbf{o}}^K(\mathbf{v}) = \exp_{\mathbf{o}}^K\left(\left[0, \mathbf{v}^E\right]\right)$$

$$= \left(\underbrace{\cosh\left(\frac{\|\mathbf{v}^E\|_2}{\sqrt{K}}\right)}_{\text{time-like dimension}}, \underbrace{\sqrt{K}\sinh\left(\frac{\|\mathbf{v}^E\|_2}{\sqrt{K}}\right)\frac{\mathbf{v}^E}{\|\mathbf{v}^E\|_2}}_{\text{space-like dimension}}\right), \tag{7}$$

where the $(,)$ denotes concatenation and the $\cdot^E$ denotes the embedding in Euclidean space .

### A.2    FORMAN-RICCI CURVATURE

Curvature is a metric used in Riemannian geometry that expresses how far a curved line deviates from a straight line, or how much a surface deviates from planarity. In this context, knowledge of the local and global geometrical features depends on an understanding of sectional curvature and Ricci curvature, respectively  Sun et al. (2024); Ye et al. (2019).

**Sectional Curvature.** This type of curvature is determined at any given point on a manifold by examining all possible two-dimensional subspaces that intersect at that point. It provides a more straightforward representation than the Riemann curvature tensor Lee (2018). Recent studies Chen et al. (2021) often treat sectional curvature uniformly across the manifold, simplifying it to a singular constant value.

**Ricci Curvature.** Ricci curvature averages the sectional curvatures at a specific point. In graph theory, various discrete versions of Ricci curvature have been developed, such as Ollivier-Ricci curvature Ollivier (2009) and Forman-Ricci curvature Forman (2003). The Ricci curvature on graphs is intended to assess how the local structure around a graph edge deviates from that of a grid graph. Notably, the Ollivier approach provides a rougher estimate of Ricci curvature, whereas the Forman method is more combinatorial and computationally efficient.

For a weighted graph $G = (V, E, w)$, the overall Forman-Ricci curvature $\overline{\mathrm{Ric}}(G)$ can be calculated as follows:

$$\overline{\mathrm{Ric}}(G) = \frac{1}{|E|} \sum_{(i,j) \in E} \mathrm{Ric}(i, j),$$

where $|E|$ represents the cardinality of the edge set $E$ (i.e., the total number of edges), and $\mathrm{Ric}(i, j)$ is the Forman-Ricci curvature of the edge $(i, j)$, computed as Southern et al. (2024)

$$\mathrm{Ric}(i, j) =: w_e \left( \frac{w_i}{w_e} + \frac{w_j}{w_e} - \sum_{e_l \sim i} \frac{w_i}{\sqrt{w_e w_{e_l}}} - \sum_{e_l \sim j} \frac{w_j}{\sqrt{w_e w_{e_l}}} \right)$$

where $w_e$ denotes the weight of the edge $e$, i.e, $(x, y)$, $w_i$ and $w_j$ are the weights of vertices $i$ and $j$, respectively. The sums over $e_l \sim k$ run over all edges $e_l$ incident on the vertex $k$ excluding $e$. Specifically, the curvature with vertex and edge weights set to 1 , is

$$\mathrm{Ric}(i, j) := 4 - d_i - d_j + 3|\#\Delta|,$$

where $d_i$ is the degree of node $i$ and $|\#\Delta|$ is the number of 3-cycles (i.e. triangles) containing the adjacent nodes.

Therefore, the overall Forman-Ricci curvature of the graph is the weighted average of the curvature values of all edges.

### A.3 LORENTZ TRANSFORMATIONS

In special relativity, Lorentz transformations are a family of linear transformations that describe the relationship between two coordinate frames in spacetime moving at a constant velocity relative to each other. They can be decomposed into a combination of a Lorentz Boost and a Lorentz Rotation Moretti (2002). The Lorentz boost, given a velocity $v \in \mathbb{R}^n$ with $\|v\| < 1$, is represented by the matrix $B$, which encodes the relative motion with constant velocity without rotation of the spatial axes. The Lorentz rotation matrix $R$ represents the rotation of spatial coordinates and is a special orthogonal matrix, i.e., $R^\top R = I$ and $\det(R) = 1$.

**Definition 5** (Lorentz Boost). *A Lorentz boost represents a change in velocity between two coordinate frames without rotation of the spatial axes. Given a velocity $\mathbf{v} \in \mathbb{R}^n$ (relative to the speed of light) with $\|\mathbf{v}\| < 1$, and the Lorentz factor $\gamma = \frac{1}{\sqrt{1 - \|\mathbf{v}\|^2}}$, the Lorentz boost matrix is defined as:*

$$\mathbf{B} = \begin{bmatrix} \gamma & -\gamma \mathbf{v}^\top \\ -\gamma \mathbf{v} & \mathbf{I} + \frac{\gamma^2}{1+\gamma} \mathbf{v}\mathbf{v}^\top \end{bmatrix} \tag{8}$$

*where $\mathbf{I}$ is the $n \times n$ identity matrix.*

A Lorentz boost describes the geometric transformation between two inertial reference frames moving at a constant relative velocity, which involves a hyperbolic rotation in the space-time plane.

**Definition 6** (Lorentz Rotation). *A Lorentz rotation describes a rotation of the spatial coordinates. The Lorentz rotation matrix is defined as:*

$$\mathbf{R} = \begin{bmatrix} 1 & \mathbf{0}^\top \\ \mathbf{0} & \tilde{\mathbf{R}} \end{bmatrix} \tag{9}$$

*where $\tilde{\mathbf{R}} \in SO(n)$ is a special orthogonal matrix satisfying $\tilde{\mathbf{R}}^\top \tilde{\mathbf{R}} = \mathbf{I}$ and $\det(\tilde{\mathbf{R}}) = 1$.*

A Lorentz rotation represents a geometric rotation or change of orientation in the spatial dimensions of the space-time manifold, while leaving the time dimension unchanged.

Both the Lorentz boost and the Lorentz rotation are linear transformations defined directly in the Lorentz model. For any point $\mathbf{x} \in \mathcal{L}_K^n$, we have $\mathbf{B}\mathbf{x} \in \mathcal{L}_K^n$ and $\mathbf{R}\mathbf{x} \in \mathcal{L}_K^n$.

### A.4 THE FEDAVG ALGORITHM

Federated Learning (FL) is a distributed learning approach that enables the training of machine learning models using data residing on local devices. A cornerstone algorithm within the FL paradigm is the FedAvg algorithm McMahan et al. (2017). FedAvg is particularly effective for scenarios where data is decentralized and not identically distributed across participants.

---

**Algorithm 1:** FedAvg

---

**Input** : Model parameters $\boldsymbol{\theta}$, learning rate $\eta$, and client dataset $\mathcal{D}_c$ for each client $c \in \mathcal{C}$
**Output** : Aggregated model parameters $\boldsymbol{\theta}$

1 Initialize model parameters $\boldsymbol{\theta}^{(0)}$;
2 **for** *each communication round $r$* **do**
3    **for** *each client $c$ in $\mathcal{C}$* **do**
4       Client $c$ receives global model parameters $\boldsymbol{\theta}^{(r)}$;
5       **for** *local epochs $e$* **do**
6          Compute gradients $\nabla\mathcal{L} = \nabla_{\boldsymbol{\theta}^{(r)}} \sum_{(\mathbf{x},\mathbf{y})\in\mathcal{D}_c} \mathcal{L}_c(f(\mathbf{x};\boldsymbol{\theta}^{(r)}),y)$;
7       **end**
8       Update local model $\boldsymbol{\theta}^{(r+1)} \leftarrow \boldsymbol{\theta}^{(r)} - \eta\nabla\mathcal{L}$;
9       Send $\boldsymbol{\theta}^{(r+1)}$ to the server;
10    **end**
11    $N = \sum_{c\in\mathcal{C}} |\mathcal{D}_c|$;
12    Server aggregates models $\boldsymbol{\theta}^{(r+1)} \leftarrow \frac{|\mathcal{D}_c|}{N} \sum_{c\in\mathcal{C}} \boldsymbol{\theta}_c^{(r+1)}$;
13 **end**

---

## B METHODOLOGY AND ANALYSIS

### B.1 STATISTICS OF FORMAN-RICCI CURVATURE IN OTHER DATASETS

We have calculated the Forman-Ricci curvature (Appendix A.2) for each client in the Cora, Photo, and ogbn-arxiv datasets, which have 10 clients each. The statistics for CiteSeer dataset are shown in Figure 2 Initialization.

### B.2 THE FlatLand ALGORITHM

This section introduces the pseudocode of our FlatLand, as shown in Algorithm 2.

### B.3 DERIVATION OF PARAMETERS DISENTANGLEMENT

The reformulated Lorentz neural network in layer $l$ is shown as

$$\mathbf{x}^{(l+1)} = \mathrm{LT}(\mathbf{x}^{(l)}; \hat{\mathbf{M}}^{(l)}) = \left( \underbrace{\sqrt{\|mx_t + \mathbf{M}\mathbf{x}_s\|^2 + K}}_{\text{time-like dimension } x_t^{(l+1)}}, \quad \underbrace{mx_t + \mathbf{M}\mathbf{x}_s}_{\text{space-like dimensions } \mathbf{x}_s^{(l+1)}} \right)^T. \tag{10}$$

The loss $\mathcal{L}_c(f(\mathbf{x}; \boldsymbol{\theta}_c, \boldsymbol{\theta}_s), y)$ of client $c$, the partial derivatives can be calculated as follows:

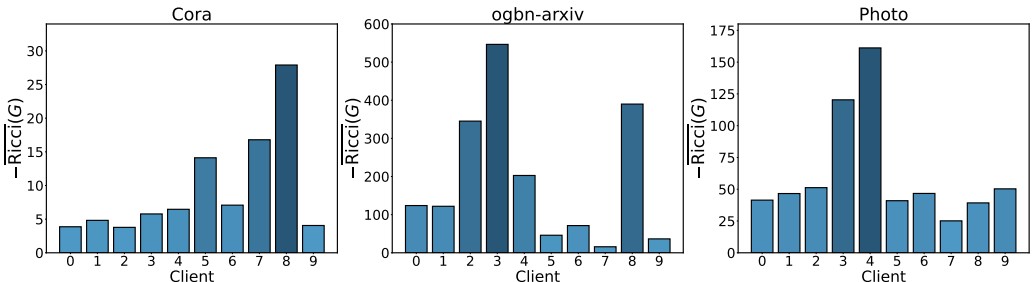

Figure 6: Averaged Forman-Ricci curvature across datasets (Cora, ogbn-arxiv, and Amazon-Photo). Higher bars indicate more pronounced non-Euclidean characteristics in these datasets.

---

**Algorithm 2:** FlatLand

---

**Input** : Personalized parameters $\boldsymbol{\theta}_c^{(0)}, K_c^{(0)}$ and dataset $\mathcal{D}_c$, for each client $c \in \mathcal{C}$
           Shared parameters $\overline{\boldsymbol{\theta}}_s^{(0)}$
           Learning rate $\eta$

**Output** : Client model parameters $\boldsymbol{\Theta}_c = \left( K_c; \boldsymbol{\theta}_c; \overline{\boldsymbol{\theta}}_s \right)$, for each client $c \in \mathcal{C}$
           Shared parameters $\overline{\boldsymbol{\theta}}_s$

1   Initialize model parameters: $\overline{\boldsymbol{\theta}}_s^{(0)}$ and $\boldsymbol{\Theta}_c^{(0)} = \left( K_c^{(0)}; \boldsymbol{\theta}_c^{(0)}; \overline{\boldsymbol{\theta}}_s^{(0)} \right)$, for $c \in \mathcal{C}$;

2   **for** *each communication round $r$* **do**

3      **for** *each client $c$ in $C$* **do**

4          $\mathbf{x} = \exp_{\mathbf{o}}^{K_c^{(r)}}(\mathbf{x})$, for $\mathbf{x} \in \mathcal{D}_c$;

5          Client $c$ receives global model parameters $\overline{\boldsymbol{\theta}}_s^{(r)}$;

6          $\boldsymbol{\Theta}_c^{(r)} = \left( K_c^{(r)}; \boldsymbol{\theta}_c^{(r)}; \overline{\boldsymbol{\theta}}_s^{(r)} \right)$ ;

7          **for** *local epochs $e$* **do**

8              Compute gradients $\nabla \mathcal{L} = \nabla_{\boldsymbol{\Theta}_c^{(r)}} \sum_{(\mathbf{x},\mathbf{y}) \in \mathcal{D}_c} \mathcal{L}_c(f(\mathbf{x}; \boldsymbol{\Theta}_c^{(r)}), y)$;

9          **end**

10         Update local model $\boldsymbol{\Theta}_c^{(r+1)} \leftarrow \boldsymbol{\Theta}_c^{(r)} - \eta \nabla \mathcal{L}$;

11         Send $\boldsymbol{\theta}_s^{(r+1)} \in \boldsymbol{\Theta}_c^{(r+1)}$ to the server;

12      **end**

13      $N = \sum_{c \in \mathcal{C}} |\mathcal{D}_c|$;

14      Server aggregates models $\overline{\boldsymbol{\theta}}_s^{(r+1)} \leftarrow \sum_{c \in \mathcal{C}} \frac{|\mathcal{D}_c|}{N} \boldsymbol{\theta}_{s_c}^{(r+1)}$;

15   **end**

---

TIME-LIKE DIMENSION $x_t^{(l+1)}$

First, we compute the partial derivative of $x_t^{(l+1)}$ with respect to the matrix $\mathbf{M}^{(l)}$ and $m^{(l)}$. Using the chain rule:

$$\frac{\partial x_t^{(l+1)}}{\partial \mathbf{M}^{(l)}} = \frac{\partial}{\partial \mathbf{M}} \sqrt{\|m^{(l)} x_t^{(l)} + \mathbf{M}^{(l)} \mathbf{x}_s^{(l)}\|^2 + K};$$

$$\frac{\partial x_t^{(l+1)}}{\partial m^{(l)}} = \frac{\partial}{\partial m} \sqrt{\|m^{(l)} x_t^{(l)} + \mathbf{M}^{(l)} \mathbf{x}_s^{(l)}\|^2 + K}.$$

Applying the chain rule, we get:

$$\begin{aligned}
\frac{\partial x_t^{(l+1)}}{\partial \mathbf{M}^{(l)}} &= \frac{1}{2} \left( \|m^{(l)} x_t^{(l)} + \mathbf{M}^{(l)} \mathbf{x}_s^{(l)}\|^2 + K \right)^{-\frac{1}{2}} \cdot 2(m^{(l)} x_t^{(l)} + \mathbf{M}^{(l)} \mathbf{x}_s^{(l)}) \cdot \frac{\partial (\mathbf{M}^{(l)} \mathbf{x}_s^{(l)})}{\partial \mathbf{M}^{(l)}} \\
&= \frac{m^{(l)} x_t^{(l)} + \mathbf{M}^{(l)} \mathbf{x}_s^{(l)}}{\sqrt{\|m^{(l)} x_t^{(l)} + \mathbf{M}^{(l)} \mathbf{x}_s^{(l)}\|^2 + K}} \cdot \frac{\partial (\mathbf{M}^{(l)} \mathbf{x}_s^{(l)})}{\partial \mathbf{M}^{(l)}}
\end{aligned} \tag{11}$$

$$\begin{aligned}
\frac{\partial x_t^{(l+1)}}{\partial m^{(l)}} &= \frac{1}{2} \left( \|m^{(l)} x_t^{(l)} + \mathbf{M}^{(l)} \mathbf{x}_s^{(l)}\|^2 + K \right)^{-\frac{1}{2}} \cdot 2(m^{(l)} x_t^{(l)} + \mathbf{M}^{(l)} \mathbf{x}_s^{(l)}) \cdot \frac{\partial (m^{(l)} \mathbf{x}_t^{(l)})}{\partial \mathbf{M}^{(l)}} \\
&= \frac{(m^{(l)} x_t^{(l)} + \mathbf{M}^{(l)} \mathbf{x}_s^{(l)})}{\sqrt{\|m^{(l)} x_t^{(l)} + \mathbf{M}^{(l)} \mathbf{x}_s^{(l)}\|^2 + K}} \cdot x_t^{(l)}
\end{aligned} \tag{12}$$

SPACE-LIKE DIMENSION $\mathbf{x}_s^{(l+1)}$

Assume that the update rule for the space-like vector $\mathbf{x}_s^{(l+1)}$ is given by the following formula:

$$\mathbf{x}_s^{(l+1)} = m^{(l)} x_t^{(l)} + \mathbf{M}^{(l)} \mathbf{x}_s^{(l)}$$

Similarly, we have

$$\frac{\partial \mathbf{x}_s^{(l+1)}}{\partial \mathbf{M}^{(l)}} = \frac{\partial \left( \mathbf{M}^{(l)} \mathbf{x}_s^{(l)} \right)}{\partial \mathbf{M}^{(l)}}, \quad \frac{\partial \mathbf{x}_s^{(l+1)}}{\partial m^{(l)}} = \frac{\partial \left( m^{(l)} \mathbf{x}_t^{(l)} \right)}{\partial m^{(l)}}. \tag{13}$$

*"Flatland"* is the space of dimension $1 : n$, serving as a metaphor for a platform where common information is exchanged and integrated. The same space-like dimension transformation $\mathbf{x}_s^{(l)} \rightarrow \mathbf{x}_s^{(l+1)}$, i.e., $\mathbf{x}_s^{(l)} \rightarrow \left( \mathbf{M}^{(l)} \mathbf{x}_s^{(l)} + m^{(l)} \mathbf{x}_s^{(l)} \right)$ in different client with different curvatures, it is easy to know that the gradient of the parameter $m$ is only related to $x_t$.

For better illustration, here, we let $\mathbf{x}^{(l)} \in \mathcal{L}_K^n$, $\mathbf{x}^{(l+1)} \in \mathcal{L}_K^n$, and $\hat{\mathbf{M}}^{(l)} \in \mathbb{R}^{(n+1) \times (n+1)}$. The introduced *"Flatland"* $\mathbb{R}^n$ is defined as a manifold spanning dimensions $1$ to $n$. This construct serves as a metaphorical platform for the exchange and integration of common information, and $x_t$ serves as the heterogeneous information. Consider the same transformation of a space-like vector $\mathbf{x}_s^{(l)}$ to $\mathbf{x}_s^{(l+1)}$ in different clients, formulated as

$$\mathbf{x}_s^{(l)} \rightarrow \left( \mathbf{M}^{(l)} \mathbf{x}_s^{(l)} + m^{(l)} \mathbf{x}_s^{(l)} \right),$$

it is easy to recognize that the gradient of the parameter $m^{(l)}$ depends solely on $x_t$ (Equation (12) and Equation (13)). Therefore, the update of parameter $m^{(l)}$ is only related to heterogeneous information and transmitted to the server side for aggregation may lead to performance degradation.

### B.4 Proof of Corollary 1

*Proof.* Let $\mathbf{x} = \begin{bmatrix} x_t \\ \mathbf{x}_s \end{bmatrix} \in \mathcal{L}_K^n$, where $x_t \in \mathbb{R}, \mathbf{x}_s \in \mathbb{R}^n$. According to Equation (4), we have:

$$\text{LT}\left(\mathbf{x}; \Phi(\hat{\mathbf{M}}, \mathbf{N})\right) = \begin{bmatrix} \sqrt{\|mx_t + \mathbf{N}\mathbf{x}_s\|^2 + K} \\ mx_t + \mathbf{N}\mathbf{x}_s \end{bmatrix}$$

We need to prove that $\text{LT}(\mathbf{x}; \Phi(\hat{\mathbf{M}}, \mathbf{N})) \in \mathcal{L}_K^m$, i.e., to prove that it satisfies the definition condition of the Lorentz manifold $\langle \cdot, \cdot \rangle_{\mathcal{L}} = -K$:

$$
\begin{aligned}
&\left\langle \text{LT}\left(\mathbf{x}; \Phi(\hat{\mathbf{M}}, \mathbf{N})\right), \text{LT}\left(\mathbf{x}; \Phi(\hat{\mathbf{M}}, \mathbf{N})\right) \right\rangle_{\mathcal{L}} \\
&= \left\langle \begin{bmatrix} \sqrt{\|mx_t + \mathbf{N}\mathbf{x}_s\|^2 + K} \\ mx_t + \mathbf{N}\mathbf{x}_s \end{bmatrix}, \begin{bmatrix} \sqrt{\|mx_t + \mathbf{N}\mathbf{x}_s\|^2 + K} \\ mx_t + \mathbf{N}\mathbf{x}_s \end{bmatrix} \right\rangle_{\mathcal{L}} \quad \text{(Definition 2)} \\
&= -\left(\sqrt{\|mx_t + \mathbf{N}\mathbf{x}_s\|^2 + K}\right)^2 + \|mx_t + \mathbf{N}\mathbf{x}_s\|^2 \\
&= -K
\end{aligned}
$$

Therefore, we have proved that $\text{LT}\left(\mathbf{x}; \Phi(\hat{\mathbf{M}}, \mathbf{N})\right) \in \mathcal{L}_K^m$. $\qquad\square$

### B.5 Convergence Analysis

FedAvg converges to the global optimum at a rate of $O(\frac{1}{T})$ for strongly convex and smooth functions and non-iid data. When the learning rate is sufficiently small, the effect of $E$ steps of local updates is similar to a step update with a larger learning rate (Li et al., 2020b).

In this section, we demonstrate that FlatLand achieves a convergence rate of $O(\frac{1}{T})$ without regularization, which is consistent with FedAvg. Furthermore, when incorporating regularization similar to FedProx (Li et al., 2020a), the convergence rate can be bounded by a constant that reflects the degree of data heterogeneity, analogous to FedProx's theoretical guarantees. This analysis confirms that our special geometric enhanced decoupling strategy maintains the overall convergence properties while addressing the challenges of heterogeneous data distribution.

To simplify the analysis, we consider each client conducts full batch gradient descent with one step. At client $c$, the objective function can be generally written as

$$\min_{\boldsymbol{\theta}_c|_{c=1}^C, \boldsymbol{\theta}_s} \mathcal{L}_c(f(\mathbf{x}^{K_c}; \boldsymbol{\theta}_c, \boldsymbol{\theta}_s), y) + \lambda \|\boldsymbol{\theta}_{s_c} - \overline{\boldsymbol{\theta}}_s\|_2^2, \tag{14}$$

where $\lambda$ is a hyperparameter, $y \in \mathcal{Y}$, $\|\boldsymbol{\theta}_{s_c} - \overline{\boldsymbol{\theta}}_s\|_2^2$ is the regularization term that prevents the locally updated model $\boldsymbol{\theta}_{s_c}$ from deviating too far from the server shared parameters $\overline{\boldsymbol{\theta}}_s$.

Let $\ell_c = \mathcal{L}_c(f(\mathbf{x}^{K_c}; \boldsymbol{\theta}_c, \boldsymbol{\theta}_s), y)$, then the global loss is taken as an average of the loss of each client: $\ell = \sum_{c \in \mathcal{C}} p_c \ell_c$, where $p_c \geq 0$ and $\sum_c p_c = 1$.

The local update is performed using vanilla gradient descent with a local learning rate $\eta$ in each client, and $\boldsymbol{\Theta}_c(r) \in \mathcal{E}$ represents the weight parameters of the client $c$ in the round $r$. Then, for global round $r$,

$$\Delta\boldsymbol{\Theta}_c^{(r)} = \boldsymbol{\Theta}_c^{(r+1)} - \boldsymbol{\Theta}_c^{(r)} = -\eta\left(\nabla\ell_c(\boldsymbol{\Theta}^{(r)}) + 2\lambda\left(\boldsymbol{\theta}_{s_c} - \hat{\boldsymbol{\theta}}_s\right)\right).$$

To better calculate the difference between personalized parameters and shared parameters, we let

$$\boldsymbol{\Theta}_c^{(r)} = \boldsymbol{\theta}_c^{(r)} + \boldsymbol{\theta}_s^{(r)}$$

, where, $\boldsymbol{\theta}_c^{(r)} = [m^{(r)} \quad \mathbf{o}]$, $\boldsymbol{\theta}_s^{(r)} = [\mathbf{o} \quad \mathbf{M}^{(r)}]$.

Specifically, the global aggregation procedure is conducted by taking the average of local updates of shared parameters $\boldsymbol{\theta_s}$ of all $|\mathcal{C}|$ clients. According to

$$\boldsymbol{\theta}_s^{(r+1)} = \bar{\boldsymbol{\theta}}_s^{(r)} = \sum_{c \in \mathcal{C}} \frac{|\mathcal{D}_c|}{N} \boldsymbol{\theta}_{s_c}^{(r)} = \sum_{c \in \mathcal{C}} p_c \boldsymbol{\theta}_{s_c}^{(r)}$$

We make the following standard Assumption commonly used in non-convex optimization (Li et al., 2020b; Reddi et al., 2020).

**Assumption 1** (L-smoothness). $\forall_{c \in \mathcal{C}} \ell_c$ *are L-smooth: for all* $\boldsymbol{\Theta}_1 \in \mathbb{E}$ *and* $\boldsymbol{\Theta}_2 \in \mathbb{E}$,

$$\ell_c(\boldsymbol{\Theta}_1) \leq \ell_c(\boldsymbol{\Theta}_2) + (\boldsymbol{\Theta}_1 - \boldsymbol{\Theta}_2)^T \nabla \ell_c(\boldsymbol{\Theta}_2) + \frac{L}{2} \|\boldsymbol{\Theta}_1 - \boldsymbol{\Theta}_2\|_2^2.$$

**Assumption 2** (Bounded Gradients). *The function* $\ell_c(\boldsymbol{\Theta})$ *have G-bounded gradients, i.e., for any* $c \in \mathcal{C}$, $\boldsymbol{\Theta} \in \mathbb{R}^d$ *we have* $\|\nabla \ell_c(\boldsymbol{\Theta})\| \leq G$.

**Lemma 1** (Smooth Decent Lemma). *Let* $\ell : \mathcal{E} \to \mathbb{R}$ *be an L-smooth function. Then for any* $\boldsymbol{\Theta}^{(r)}, \boldsymbol{\Theta}^{(r+1)} \in \mathbb{E}$, *the following inequality holds:*

$$\ell(\boldsymbol{\Theta}^{(r+1)}) \leq \ell(\boldsymbol{\Theta}^{(r)}) + \langle \nabla \ell(\boldsymbol{\Theta}^{(r)}), \Delta \boldsymbol{\Theta}^{(r)} \rangle + \frac{L}{2} \|\Delta \boldsymbol{\Theta}^{(r)}\|^2.$$

Let $\delta^{(r)} = 2\lambda \sum_{c \in \mathcal{C}} \frac{|\mathcal{D}_c|}{N} \left( \boldsymbol{\theta}_{s_c} - \bar{\boldsymbol{\theta}}_{\boldsymbol{s}} \right)$. Based on Lemma 1, we have

$$
\begin{aligned}
\ell(\boldsymbol{\Theta}^{(r+1)}) &\leq \ell(\boldsymbol{\Theta}^{(r)}) + \langle \nabla \ell(\boldsymbol{\Theta}^{(r)}), \Delta \boldsymbol{\Theta}^{(r)} \rangle + \frac{L}{2} \|\Delta \boldsymbol{\Theta}^{(r)}\|^2 \\
&= \ell(\boldsymbol{\Theta}^{(r)}) + \left\langle \nabla \ell(\boldsymbol{\Theta}^{(r)}), -\eta \left( \nabla \ell(\boldsymbol{\Theta}^{(r)}) + \delta^{(r)} \right) \right\rangle + \frac{L\eta^2}{2} \|\nabla \ell(\boldsymbol{\Theta}^{(r)}) + \delta^{(r)}\|^2 \\
&= \ell(\boldsymbol{\Theta}^{(r)}) - \eta \left\langle \nabla \ell(\boldsymbol{\Theta}^{(r)}), \nabla \ell(\boldsymbol{\Theta}^{(r)}) + \delta^{(r)} \right\rangle + \frac{L\eta^2}{2} \|\nabla \ell(\boldsymbol{\Theta}^{(r)}) + \delta^{(r)}\|^2 \\
&= \ell(\boldsymbol{\Theta}^{(r)}) - \eta \|\nabla \ell(\boldsymbol{\Theta}^{(r)})\|^2 - \eta \left\langle \nabla \ell(\boldsymbol{\Theta}^{(r)}), \delta^{(r)} \right\rangle + \frac{L\eta^2}{2} \|\nabla \ell(\boldsymbol{\Theta}^{(r)})\|^2 + L\eta^2 \langle \nabla \ell(\boldsymbol{\Theta}^{(r)}, \delta^{(r)}) \rangle + \frac{L\eta^2}{2} \|\delta^{(r)}\|^2 \\
&= \ell(\boldsymbol{\Theta}^{(r)}) + (\frac{L\eta^2}{2} - \eta) \|\nabla \ell(\boldsymbol{\Theta}^{(r)})\|^2 + \frac{L\eta^2}{2} \|\delta^{(r)}\|^2 + (L\eta^2 - \eta) \left\langle \nabla \ell(\boldsymbol{\Theta}^{(r)}), \delta^{(r)} \right\rangle \\
&= \ell(\boldsymbol{\Theta}^{(r)}) + (\frac{L\eta^2}{2} - \eta) \|\nabla \ell(\boldsymbol{\Theta}^{(r)})\|^2 + \frac{L\eta^2}{2} \|\delta^{(r)}\|^2 + \frac{L\eta^2 - \eta}{2} \left( \|\nabla \ell(\boldsymbol{\Theta}^{(r)})\|^2 + \|\delta^{(r)}\|^2 - \|\nabla \ell(\boldsymbol{\Theta}^{(r)}) + \delta^{(r)}\|^2 \right) \\
&= \ell(\boldsymbol{\Theta}^{(r)}) + (L\eta^2 - \frac{3\eta}{2}) \|\nabla \ell(\boldsymbol{\Theta}^{(r)})\|^2 + (L\eta^2 - \frac{\eta}{2}) \|\delta^{(r)}\|^2 - \frac{L\eta^2 - \eta}{2} \|\nabla \ell(\boldsymbol{\Theta}^{(r)}) + \delta^{(r)}\|^2
\end{aligned}
\tag{15}
$$

We select $\eta = \frac{1}{L}$, so we we have

$$\ell(\boldsymbol{\Theta}^{(r+1)}) \leq \ell(\boldsymbol{\Theta}^{(r)}) - \frac{1}{2L} \|\nabla \ell(\boldsymbol{\Theta}^{(r)})\|^2 + \frac{1}{2L} \|\delta^{(r)}\|^2 \tag{16}$$

Rearrange the above inequality and we have

$$\|\nabla \ell(\boldsymbol{\Theta}^{(r)})\|^2 \leq 2L \left( \ell(\boldsymbol{\Theta}^{(r+1)}) - \ell(\boldsymbol{\Theta}^{(r)}) \right) + \|\delta^{(r)}\|^2 \tag{17}$$

Then, sum $r$ from 1 to $T$, we have

$$\min_{r \in [T]} \|\nabla \ell(\boldsymbol{\Theta}^{(r)})\| \leq \frac{2L \left( \ell(\boldsymbol{\Theta}^{(r+1)}) - \ell(\boldsymbol{\Theta}^{(r)}) \right)}{T} + \frac{1}{T} \sum_{r \in [T]} \|\delta^{(r)}\|^2 \tag{18}$$

**Definition 7** ($B$-local dissimilarity). *The local functions* $\ell_c$ *are B-locally dissimilar at* $\boldsymbol{\Theta}$ *if*
$$\mathbb{E}_c[\|\nabla \ell_c(\boldsymbol{\Theta})\|^2] \leq \|\nabla \ell(\boldsymbol{\Theta})\|^2 B^2.$$

*We further define* $B(\boldsymbol{\Theta}) = \sqrt{\frac{\mathbb{E}_c[\|\nabla \ell_c(\boldsymbol{\Theta})\|^2]}{\|\nabla \ell(\boldsymbol{\Theta})\|^2}}$ *for* $\|\nabla \ell(\boldsymbol{\Theta})\| \neq 0$.

**Definition 8** ($\gamma$-inexact solution). *For a function $h(w; w_0) = F(w) + \lambda\|w - w_0\|^2$, and $\gamma \in [0, 1]$, we say $w^*$ is a $\gamma$-inexact solution of $\min_w h(w; w_0)$ if $\|\nabla h(w^*; w_0)\| \leq \gamma \|\nabla h(w_0; w_0)\|$, where $\nabla h(w; w_0) = \nabla F(w) + \mu(w - w_0)$, where, $\mu = 2\lambda$. Note that smaller $\gamma$ corresponds to higher accuracy.*

Using the notion of $\gamma$-inexactness for each local client, we can define $e_c^{(r)}$ such that

$$\nabla \ell_c \left( \boldsymbol{\Theta}_c^{(r+1)} \right) + \mu \left( \hat{\boldsymbol{\theta}}_s^{(r)} - \boldsymbol{\theta}_{s_c}^{(r)} \right) + \mu \left( \boldsymbol{\theta}_c^{(r+1)} - \boldsymbol{\theta}_c^{(r)} \right) - e_c^{(r)} = 0,$$
$$\|e_c^{(r)}\| \leq \gamma \|\nabla \ell_c \left( \boldsymbol{\Theta}_c^{(r)} \right)\|. \tag{19}$$

Then we have

$$\boldsymbol{\theta}_s^{(r+1)} - \boldsymbol{\theta}_s^{(r)} = \frac{-1}{\mu}\mathbb{E}_c \left[ \nabla \ell_c \left( \boldsymbol{\Theta}_c^{(r)} \right) \right] + \frac{1}{\mu}\mathbb{E}_c[e_c^{(r)}] - \mathbb{E}_c \left[ \Delta \boldsymbol{\theta}_c^{(r)} \right], \tag{20}$$

According to (Li et al., 2020a) and triangle inequality, when a regularization is incorporated, ($\lambda > 0$), we have

$$\frac{1}{4\lambda^2}\|\delta^{(r)}\|^2 \leq \left( \mathbb{E}_c \left[ \|\boldsymbol{\theta}_s^{(r+1)} - \boldsymbol{\theta}_{s_c}^{(r)}\| \right] \right)^2 \leq \left( \frac{1+\gamma}{\bar{\mu}} \right)^2 \left( \mathbb{E}_c \left[ \|\nabla \ell_c \left( \boldsymbol{\Theta}_c^{(r)} \right) - \Delta \boldsymbol{\theta}_c^{(r)}\| \right] \right)^2$$

$$\leq \left( \frac{1+\gamma}{\bar{\mu}} \right)^2 \left( \mathbb{E}_c \left[ \|\nabla \ell_c \left( \boldsymbol{\Theta}_c^{(r)} \right) - \Delta \boldsymbol{\theta}_c^{(r)}\|^2 \right] \right)$$

$$\leq \frac{B^2(1+\gamma)^2}{\bar{\mu}^2}\mathbb{E} \left[ \|\nabla \ell_c \left( \boldsymbol{\Theta}_c^{(r)} \right)\|^2 \right] + C,$$

Based on the assumption of the bounded gradients (Assumption 2), we find that the $\delta^{(r)}$ is also bounded. Specifically, $C = \left( \frac{1+\gamma}{\bar{\mu}} \right)^2 \mathbb{E}_c[\|\Delta\boldsymbol{\theta}_c\|^2] \approx \left( \frac{1+\gamma}{\bar{\mu}} \right)^2 \mathbb{E}[\|\Delta M_c\|^2]$. $\|\delta^{(r)}\|^2$ measures the degree of data heterogeneity.

Overall, when $\lambda = 0$, the term $\delta^{(r)} = 0$, eliminating the impact of data heterogeneity and resulting in a convergence rate of $O\left(\frac{1}{T}\right)$, consistent with FedAvg. And when incorporating regularization ($\lambda > 0$), we establish that $\left\|\delta^{(r)}\right\|^2$ is bounded, analogous to the theoretical guarantees provided by FedProx (Li et al., 2020a)..

### B.6 TIME AND SPACE COMPLEXITY COMPARED WITH FEDAVG

We analyze the computational complexity of FlatLand compared to FedAvg, which gives insight for the scalability.

**Local Update** The additional operations in FlatLand's local update phase compared with FedAvg - curvature estimation (Section 5.1), exponential map (line 4 in Algorithm 2, Equation 7). Notably, the curvature estimation can be *pre-computed* since each client's data distribution corresponds to a constant curvature value. For exponential map, the transformation only requires *a single* non-linear mapping operation based on the norm of input samples with the time complexity of $O(1)$. These norms can also be *pre-computed and cached*. Therefore, while FlatLand introduces these additional steps compared to FedAvg, their practical computational overhead is limited due to pre-computation opportunities and constant-time operations.

**Aggregation** FlatLand and FedAvg have the same aggregation time complexity when the hidden embedding dimension is the same. Though FlatLand introduces extra time-like space parameters, it only aggregates shared parameters $\boldsymbol{\theta}_s$ while maintaining personalized parameters. The overhead of the shared parameters is the same. Moreover, FlatLand can perform better in low dimensionality (Section 7.3), which potentially reduces practical communication costs.

**Space Requirements and Storage** FlatLand requires extra $O(d + 1)$ storage per client compared to FedAvg due to the additional time-like dimension and curvature parameter, where $d$ is the hidden dimension. Since typically $d$ is small, the increase in storage is small. Moreover, FlatLand

Table 4: Statistics of node classification datasets. We report the (average) number of nodes, edges, classes, clustering coefficient, and heterogeneity for different numbers of clients.

| Dataset | Cora | | | Citeseer | | | ogbn-arxiv | | | Amazon-Photo | | |
|---|---|---|---|---|---|---|---|---|---|---|---|---|
| # Clients | 1 | 10 | 20 | 1 | 10 | 20 | 1 | 10 | 20 | 1 | 10 | 20 |
| # Classes | 7 | | | 6 | | | 40 | | | 8 | | |
| Avg. # Nodes | 2,485 | 249 | 124 | 2,120 | 212 | 106 | 169,343 | 16,934 | 8,467 | 7,487 | 749 | 374 |
| Avg. # Edges | 10,138 | 891 | 422 | 7,358 | 675 | 326 | 2,315,598 | 182,226 | 86,755 | 238,086 | 19,322 | 8,547 |
| Avg. Clustering Coefficient | 0.238 | 0.259 | 0.263 | 0.170 | 0.178 | 0.180 | 0.226 | 0.259 | 0.269 | 0.410 | 0.457 | 0.477 |
| Heterogeneity | N/A | 0.606 | 0.665 | N/A | 0.541 | 0.568 | N/A | 0.615 | 0.637 | N/A | 0.681 | 0.751 |

Table 5: Statistics of graph classification datasets. We report the (average) number of graphs, nodes, edges, classes, and node features of each dataset.

| Dataset | CHEM | | | | | | | BIO | | | SN | | |
|---|---|---|---|---|---|---|---|---|---|---|---|---|---|
| | MUTAG | BZR | COX2 | DHFR | PTC_MR | AIDS | NCI1 | ENZYMES | DD | PROTEINS | COLLAB | IMDB-BINARY | IMDB-MULTI |
| # Graphs | 188 | 405 | 467 | 467 | 344 | 2000 | 4110 | 600 | 1178 | 1113 | 5000 | 1000 | 1500 |
| Avg. # Nodes | 17.93 | 35.75 | 41.22 | 42.43 | 14.29 | 15.69 | 29.87 | 32.63 | 284.32 | 39.06 | 74.49 | 19.77 | 13.00 |
| Avg. # Edges | 19.79 | 38.36 | 43.45 | 44.54 | 14.69 | 16.20 | 32.30 | 62.14 | 715.66 | 72.82 | 2457.78 | 96.53 | 65.94 |
| # Classes | 2 | 2 | 2 | 2 | 2 | 2 | 2 | 6 | 2 | 2 | 3 | 2 | 3 |
| Node Features | original | original | original | original | original | original | original | original | original | original | degree | degree | degree |

demonstrates superior performance even in low-dimensional settings compared with the Euclidean counterparts, which further limits the practical storage overhead.

This analysis suggests that FlatLand can balance the trade-off between computational overhead and model effectiveness, showing the scalability for the increase in clients. While it introduces additional operations in local computations, these overheads are limited and offer significant optimization opportunities through pre-computation and caching strategies. The method compensates for these minimal costs through reduced communication overhead and enhanced representation capabilities in the Lorentz space, making it a practical and efficient choice for personalized federated learning applications.

## C  EXPERIMENTAL SUPPLEMENTARY

### C.1  DATASETS

For federated node classification, we adopt four benchmark datasets constructed by Baek et al. (2023): Cora, CiteSeer, ogbn-arxiv, and Photo Sen et al. (2008); Hu et al. (2020); Shchur et al. (2018). Cora, CiteSeer, and ogbn-arxiv are citation graphs. Photo is a product graph. Each graph dataset is divided into a certain number of disjoint subgraphs using the METIS graph partitioning algorithm Karypis & Kumar (1995), where each subgraph belongs to an FL client. Statistics of datasets are summarized in Table 4.

For federated graph classification, we consider the non-IID settings proposed by Xie et al. (2021). In total, there are 13 graph classification datasets from three different domains, including small molecules (MUTAG, BZR, COX2, DHFR, PTC_MR, AIDS, NCI1) denoted as CHEM, bioinformatics (ENZYMES, DD, PROTEINS) denoted as BIO, and social networks (COLLAB, IMDB-BINARY, IMDB-MULTI) Morris et al. (2020) denoted as SN. To simulate data heterogeneity, three non-IID settings are constructed: (1) a cross-dataset setting based on the small molecule datasets (CHEM), (2) a cross-domain setting based on all datasets (BIO-CHEM-SN). In each setting, one dataset corresponds to one FL client. Statistics of datasets are summarized in Table 5.

### C.2  IMPLEMENTATION DETAILS

**Implementation of learnable curvature.**  $K$ is a learnable scalar parameter. To ensure the curvature remains negative (as required for hyperbolic space), we implement it as sigmoid($K$) + 0.5. This design also keeps curvature $-K$ within an effective range of $[0.5, 1.5]$, which prior work has shown to be ideal for hyperbolic models (Chen et al., 2021). Additionally, this approach maintains numerical stability while satisfying the need for a heterogeneous space.

**Implementation of node classification / graph classification task.**  For the node classification task, we employ 2-layer GCN Kipf & Welling (2017) for Euclidean models, 2-layer LGCN Chen et al.

| # clients ($\beta$) | MNIST (Acc%) 20(0.1) | MNIST (AUC%) 20(0.1) | MNIST (Acc%) 100(0.1) | MNIST (AUC%) 100(0.1) |
|---|---|---|---|---|
| FedAvg | $87.86 \pm 0.0816$ | $97.77 \pm 0.0149$ | $86.14 \pm 0.2066$ | $96.57 \pm 0.0508$ |
| FedProx | $87.53 \pm 0.0771$ | $98.81 \pm 0.0110$ | $84.50 \pm 0.1658$ | $98.22 \pm 0.0442$ |
| Ditto | $97.85 \pm 0.0191$ | $\underline{99.92} \pm 0.0012$ | $96.45 \pm 0.0415$ | $\underline{99.78} \pm 0.0047$ |
| GPFL | $92.90 \pm 0.0724$ | $99.48 \pm 0.0110$ | $96.52 \pm 0.0462$ | $99.70 \pm 0.0136$ |
| FedRep | $\underline{98.14} \pm 0.0196$ | $99.85 \pm 0.0196$ | $96.54 \pm 0.0750$ | $99.67 \pm 0.0190$ |
| FedCAC | $97.85 \pm 0.0189$ | $\underline{99.92} \pm 0.0012$ | $\underline{96.59} \pm 0.0505$ | $\mathbf{99.81} \pm 0.0080$ |
| FlatLand | $\mathbf{98.35} \pm 0.0136$ | $\mathbf{99.93} \pm 0.0011$ | $\mathbf{96.64} \pm 0.0495$ | $99.70 \pm 0.0116$ |

Table 6: Performance comparison on MNIST dataset.

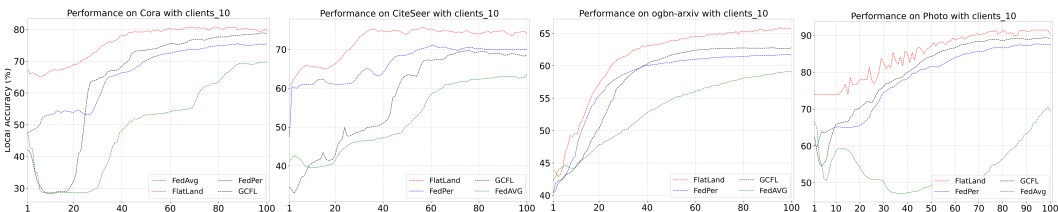

Figure 7: The convergence curves of our proposed methods and the strong baselines.

(2021) for FlatLand, and HGCN with node selection for FedHGCN Du et al. (2024). LGCN serves as the backbone for our graph learning framework, combining Lorentz linear layers (Equation 2) with graph aggregation operations, similar to how Euclidean counterparts like GCN and GIN integrate linear layers with graph aggregation. Each layer applies a Lorentz transformation followed by neighbor aggregation using the adjacency matrix to get the node representations. We conduct 100 rounds for Cora/CiteSeer and 200 rounds for larger datasets like Photo/ogbn-arxiv, with 1-3 local epochs, use 128-dim hidden layers. For graph classification, we use 3-layer GIN Xu et al. (2018) as the Euclidean encoder, and the same 3-layer hyperbolic encoders as node classification for hyperbolic models, with 1 local epoch and 200 rounds. The learning rate is chosen from $\{0.01, 0.001\}$, and weight decay uses $1e-5$. We optimize with Adam, and calculate node-level / graph-level accuracy averaged across clients. All experiments are implemented in Python3.10, PyTorch, and run on an RTX A6000 GPU, 40G storage. Each client is allocated a worker with one round of around 1 second for one epoch in the node classification task.

## C.3 EXPERIMENTS ON IMAGE DATASETS

In this section, we evaluate the effectiveness of our proposed method, FlatLand, on the MNIST dataset to demonstrate its performance on image data. We compare our method with several baseline algorithms in the context of personalized federated learning (PFL). The experiments are designed to assess the performance under different numbers of clients and to emphasize data heterogeneity.

We conducted experiments on the MNIST dataset to validate the effectiveness of our proposed method, FlatLand, on image data. The dataset was partitioned among clients using a Dirichlet distribution with a concentration parameter $\beta = 0.1$, introducing high data heterogeneity to simulate non-i.i.d. scenarios common in federated learning. We compared FlatLand against several baseline methods — FedAvg (McMahan et al., 2017), FedProx (Li et al., 2020a), Ditto Li et al. (2021a), GPFL (Zhang et al., 2023a), FedRep (Collins et al., 2021), and FedCAC (Wu et al., 2023) — under two settings with 20 and 100 clients. All experiments were implemented using PFLib (Zhang et al., 2023c).

These results demonstrate that FlatLand performs competitively on image data. This indicates that FlatLand effectively handles high data heterogeneity and scales well with different numbers of clients. Besides, the significant performance gap between FlatLand and traditional federated learning methods like FedAvg and FedProx highlights the effectiveness of our approach in highly heterogeneous settings.

### C.4    PARIAL PARTICIPATION RATE

We conducted extensive experiments with an increased number of clients (50 clients) in the Cora dataset, which represents a large client pool configuration in graph federated learning scenarios (Du et al., 2024). The results demonstrate that our method maintains its effectiveness even with an expanded client base. Furthermore, we investigated the impact of partial client participation, where only a fraction of clients participate in each aggregation round. Figure 8 illustrates the performance comparison between FedAvg and FlatLand under different participation rates on the Cora dataset with 50 clients.

The experimental results show that FlatLand exhibits remarkable robustness across various participation rates. Even with only 10% client participation (5 clients), FlatLand achieves an accuracy of 81.82%, while FedAvg only reaches 18.14%. As the participation rate increases, FlatLand maintains consistently high performance. In contrast, FedAvg shows performance fluctuations.

These findings confirm that FlatLand can maintain high performance even under low client participation scenarios, demonstrating its practical value for real-world federated learning applications where full client participation may not always be feasible. The robust performance under partial participation is particularly important for federated learning systems, where coordinating all clients simultaneously can be challenging.

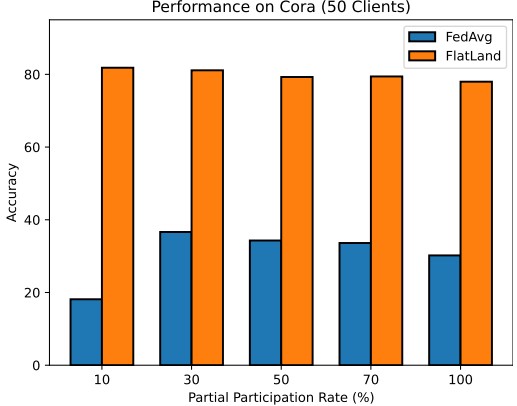

Figure 8: Performance comparison between FedAvg and FlatLand under different client participation rates on Cora dataset with 50 clients.

### C.5    CONVERGENCE CURVES

The convergence curves are shown in Figure 7. As the figures demonstrate, our proposed method can achieve better convergence speed, highlighting the superiority of our proposed approach.

### C.6    BROADER IMPACTS

Our personalized federated learning method is a major advancement for privacy-preserving, trustworthy AI. Enabling collaborative training of highly personalized models without compromising data privacy enhances user privacy protection and fosters broader adoption of ethical personalized AI technologies. Crucially, it improves personalized user experiences through accurate, tailored services while actively building transparent, user-centric personalized AI systems to boost public trust. Potential risks can be mitigated through robust safeguards, vigilance, and stakeholder collaboration.

