# OpenReview forum: "Personalized Federated Learning via Tailored Lorentz Space"
_ICLR.cc/2025/Conference — Submitted to ICLR 2025_

### Official Review · Reviewer_KA7L · 2024-10-23

**Soundness:** 2
**Presentation:** 2
**Contribution:** 1
**Rating:** 3
**Confidence:** 4

**Summary:**

The paper proposes a personalized FL method called Flatland. Flatland follows a standard Federated Averaging framework. Each client $c$ first embeds their data into a Lorentz space with a personalized (learned) curvature coefficient $K_c$. They then run local training of a Lorentz model (neural network consisting of so called Lorentz layers that are designed so that if the inputs are in a Lorentz space then so are the outputs). The parameters of this network that correspond to the first dimension of the embedded data are personalized, per client, parameters, while the remaining parameters are shared. The shared parameters are updated using FedAvg, the personalized parameters are kept on the client.

The paper shows that the specific decomposition proposed still maintained the Lorentz property of the network. The paper then carries out an empirical evaluation of the method on several different graph datasets, for both node and graph classification tasks. It compares to a range of baselines.

**Strengths:**

* The proposed method is easy to integrate into standard FL training pipelines. It trains using federated averaging and aggregates client updates, meaning it is compatible with privacy preserving techniques such as secure aggregation and differential privacy.

* The originality of the paper is that it is the first to propose using a different Lorentz embedding for each client in order to better capture the differing client data distributions.

* The experiments demonstrate good performance on node classification in real world datasets with small numbers of clients

* The authors provide ablation studies examining the impact of different components of the method

**Weaknesses:**

I do not believe the paper makes a significant enough contribution, either theoretically or empirically:

Regarding theory, there is a lack of any substantial theoretical contributions

* As far as I can tell Proposition 1 and Corollary 1 are both instant consequences of the definition of a Lorentz network (eq 2), giving a name to the first row and column of the matrix does not require reproving anything. Even more confusing is Corollary 1 and it’s proof, Proposition 1 states that $\forall M, LT(x;M) \in \mathcal{L}$ then of course it is also true that $LT(x;\Phi(M, N)) \in \mathcal{L}$ because it holds for all matrices in the second entry, this does not require a proof as given in the appendix.
* The remainder of section 6 makes some hand wavy and confusing claims and does not prove any concrete properties or statements about the method.
* There are many potentially interesting directions here the authors could have taken, e.g. what is the convergence rate, how does the lorentz embedding impact convergence and/or performance, what types of distribution heterogeneity are well modeled by the Lorentz space etc.

Regarding empirical contribution. My overall concern here centers around the empirical evaluation being too narrow. Essentially the evaluation is limited to federated cross-silo, graph datasets, which to me makes it a specialized method rather than a general one, as the authors claim.

* The empirical evaluation is only on graph datasets. The authors present the method as a general personalized federated algorithm and claim that the method is applicable to other data modalities; they present no evidence of this. I am also not convinced that this would be the case, state of the art methods that deal with the most common data modalites (text, image, video, tabular) do not make use of Lorentz spaces.

* The number of clients in each dataset is very low. Therefore, the experiments do not provide evidence that the method works in a cross-device setting. Crucially, the authors do not mention the cohort size they used in federated averaging, and given the low number of clients in their experiments I am assuming that all clients participate in each round of training. I have concerns about if the method would even work without full client participation in each round (if client A has participated before and client B has not, then A will have already updated their personalized parameters often, while B is starting from initialization, this is going to lead to very heterogeneous shared parameter updates)

* I believe the authors need to provide experiments for other data modalities, with larger numbers of clients as well as partial client participation

There are some minor typos, e.g. I think in section 5.2 $m^{(l)}$ should be in $\mathbb{R}^m$, not $\mathbb{R}^{m+1}$ and $M^{(l)}$ should be in $\mathbb{R}^{m\times n}$. Also references to tables in section 7 are off and need to be checked e.g. on line 452, Table 3 should be Table 1.

Overall, I would need to see substantial improvements to the theoretical and/or empirical contributions, as suggested in the final bullet points of each section above.

**Questions:**

* Why do the $v$ parameters not appear anywhere in Equation (4)?
* How does Lorentz space actually model heterogeneity? The paper seems to claim that $x_t$ captures the heterogeneity between client distributions, while the client distributions in Flatland i.e. $x_S$ should be the same. This is an interesting idea but no evidence is provided. I would like to see either theoretical or empirical evidence of this fact.
* Why do you decouple the parameters the way you do? This is related to the above point. It seems that the parameters are decoupled this way so that the heterogeneous feature $x_t$ is fed into the personalized portion of the network $m$ in Equation (4), while the homogeneous features $x_S$ are fed into the shared portion of the network, $M$ in Equation (4). However, as far as I can tell this only works for the first layer since the output of layer 1 $m x_t + M^{(1)} x_S$ will be fed into the shared part of layer 2 $M^{(2)}$ meaning that the updates to shared parameters from layer 2 onwards are still impacted by heterogeneous parts of the client data.
* Does the method work for larger numbers of clients, without full client participation each round?
* What do you mean by statistically significant for the results in Table 1? How can you say that FlatLand outperforms GCFL in graph classification when the mean accuracies are very close with very large overlap of the standard deviation intervals. This looks to me like noise rather than signal.
* It is not clearly defined what the embedding dimension is in Section 7.3. Is this the hidden dimension of the model?

---

> ### Author Response · Authors · 2024-11-22
> **Response (1/4)**
>
> Thank you for your valuable comments and suggestions. Below, we will summarize the concerns with "**C:**" and our responses with "**R:**"
>
> &nbsp;
>
> ---
>
> **C1: Regarding the theoretical part**
>
> **R:** Thanks for your comments.  To clarify, this theoretical analysis is not clamied as our main contribution, it serves to establish the mathematical rigor of our novel personalized federated learning strategy in hyperbolic space. Let us elaborate on the importance of our analysis
>
>
> 1. Corollary 1 is important because we are the first to apply Lorentz Neural Networks to the federated learning setting. This means that after each aggregation step, the parameters are updated, changing the model. The $\mathbf{N}$ in our case should be $\mathbf{M}_c - \eta \sum \frac{|\mathcal{D}_c|}{N} \nabla \mathbf{M}_c$, where $\nabla \mathbf{M}_c$ is the gradient of the shared parameters $\mathbf{M}$, $N$ is the total number of samples. Due to hyperbolic space's non-Euclidean nature, parameter changes can directly affect the validity of output representations. Thus, ensuring output representations remain in the hyperbolic space after parameter updates is **important for the correctness of hyperbolic neural networks, an issue unique to this setting.** While not claimed as a core contribution, the proof of Corollary 1 is essential to address this challenge specific to the federated learning of hyperbolic models.
>
> 2. The importance of other analysis: The goal of the remaining part is to show the rationale behind our proposed method from different perspectives, as stated in the paper. Our emphasis is on analyzing our approach from the perspective of hyperbolic geometry. For example, we perform a non-linear debiasing operation on the input data. Moreover, we **provide interpretable insights** drawing an analogy from special relativity, treating each client as a distinct space-time.
>
> In the paper, we have added an analysis of the convergence analysis of our proposed method in Appendix B.5. We show that the decoupling strategy does not impede the convergence of the model. In future work, we plan to further investigate the specific impact of the hyperbolic curvature on model performance, such as its effect on generalization error and other metrics. We believe this would be an interesting theoretical research direction about how geometry can directly influence the models’ performance in various tasks.

---

> ### Author Response · Authors · 2024-11-22
> **Response (2/4)**
>
> **C2: Regarding the experiments part**
>
> **R:** We appreciate the reviewer's comments on our empirical evaluation. We would like to clarify several important points:
>
> &nbsp;
>
> **1.The reason we focus on graph data:**
>
> - Our focus on graph data is theoretically motivated. Graphs inherently exhibit non-Euclidean structures and significant heterogeneity, as demonstrated in previous theoretical analyses [1-2].
> - More importantly, there is a **theoretical correlation between the heterogeneity of graph distributions and hyperbolic curvature** [1]. In contrast, for other domains, such as images or text, hyperbolic geometry has been studied more thoroughly in graph data, where explicit complex structures of heterogeneity are present, serving as an excellent testbed for interpreting how hyperbolic geometry handles hierarchical relationships and heterogeneous structures. Therefore, it provides a clear theoretical foundation for validating our method's effectiveness in handling complex geometric properties.
>
> &nbsp;
>
> **2. Applicability to other domains and settings:**
>
> We further conducted experiments with **an increased number of clients** (50 clients) in the Cora dataset, which represents a large client pool configuration of federated learning scenarios in graph datasets. The results demonstrate that our method maintains its effectiveness even with an expanded client base.  The below table illustrates the performance comparison between FedAvg and FlatLand under **different participation rates** **on the Cora dataset with 50 clients**. (We added the results in Appendix C4 in revised paper).
>
>
> | Participate Rate |  0.1  |  0.3  |  0.5  |  0.7  | **1.0** |
> | :--------------: | :---: | :---: | :---: | :---: | :-----: |
> |      FedAvg      | 18.14 | 36.64 | 34.30 | 33.61 |  30.20  |
> |   **FlatLand**   | **81.82** | **81.11** | **79.27** | **79.42** |  **77.98**  |
>
>
> **The experimental results show that FlatLand exhibits remarkable robustness in a larger number of clients and across various participation rates,** even with only 10\% client participation (5 clients). These findings confirm that FlatLand can maintain high performance even under low client participation scenarios, demonstrating its practical value federated learning applications where full client participation may not always be feasible.
>
> &nbsp;
>
>
> Our method can be readily transferred to other fields that use linear neural networks.  In the following, we provide additional experiments on widely used **image dataset**, MNIST (also considering a larger number of clients). The results are promising compared with the strong baselines [3-5]:
>
>
> | **Dataset** | **#Clients** | **FedAvg (\%)** | **FedProx (\%)** | **Ditto (\%)** | **GPFL (\%)** | **FedRep (\%)** | **FedCAC (\%)** | **FlatLand (\%)** |
> | --- | --- | --- | --- | --- | --- | --- | --- | --- |
> | MNIST (Acc) | 20 | $87.86 \pm 0.0816$ | $87.53 \pm 0.0771$ | $97.85 \pm 0.0191$ | $92.90 \pm 0.0724$ | $98.14 \pm 0.0196$ | $97.85 \pm 0.0189$ | $\mathbf{98.35} \pm 0.0136$ |
> | MNIST (AUC) | 20 | $97.77 \pm 0.0149$ | $98.81 \pm 0.0110$ | $99.92 \pm 0.0012$ | $99.48 \pm 0.0110$ | $99.85 \pm 0.0196$ | $99.92 \pm 0.0012$ | $\mathbf{99.93} \pm 0.0011$ |
> | MNIST (Acc) | 100 | $86.14 \pm 0.2066$ | $84.50 \pm 0.1658$ | $96.45 \pm 0.0415$ | $96.52 \pm 0.0462$ | $96.54 \pm 0.0750$ | $96.59 \pm 0.0505$ | $\mathbf{96.64} \pm 0.0495$ |
> | MNIST (AUC) | 100 | $96.57 \pm 0.0508$ | $98.22 \pm 0.0442$ | $99.78 \pm 0.0047$ | $99.70 \pm 0.0136$ | $99.67 \pm 0.0190$ | $\mathbf{99.81} \pm 0.0012$ | $99.70 \pm 0.0116$ |
>
>
> We also conduct additional experiments with varying client participation rates from [0.1-0.7] based on the above setting with 100 clients, and FlatLand still works in this setting.
>
> | Participation Rate | 0.1 | 0.3 | 0.5 | 0.7 |
> | --- | --- | --- | --- | --- |
> | FedAvg | 85.34 ± 0.08 | 87.25 ± 0.07 | 86.60 ± 0.08 | 87.22 ± 0.07 |
> | FlatLand | 93.71 ± 0.18 | 94.40 ± 0.12 | 95.62 ± 0.11 | 96.12 ± 0.07 |
>
> &nbsp;
>
> It shows that our method can also work and perform well without full client participation in each round. Note that our method is orthogonal and non-conflicting with methods that utilize partial strategies, allowing for complementary integration with such approaches. This is NOT the main focus of this paper.
>
> &nbsp;
>
> ---
>
> **References:**
>
> [1] Krioukov, Dmitri, et al. "Hyperbolic geometry of complex networks." Physical Review E—Statistical, Nonlinear, and Soft Matter Physics 82.3 (2010): 036106.
>
> [2] SINCERE: Sequential Interaction Networks representation learning on Co-Evolving RiEmannian manifolds, WebConf 2023
>
> [3] Exploiting shared representations for personalized federated learning, ICML 2021
>
> [4] Bold but cautious: Unlocking the potential of personalized federated learning through cautiously aggressive collaboration, ICCV 2023
>
> [5] Gpfl: Simultaneously learning global and personalized feature information for personalized federated learning, ICCV 2023

---

> > ### Comment · Reviewer_KA7L · 2024-11-25
> > **Question about Participation Rate experiments**
> >
> > Could you confirm what metric the Participation Rate results tables are showing? Is it classification accuracy? Also why are you quoting AUC and how is it computed for MNIST, given that it isn't a binary classification dataset? Could you also explain why the other baselines are not present in this Participation Rate table?

---

> > > ### Author Response · Authors · 2024-11-27
> > > **Supplementary of Response (2/4)**
> > >
> > > Dear Reviewer KA7L,
> > >
> > > &nbsp;
> > >
> > > Due to space limitations in Response 2/4, we include the updated supplementary experimental results of the partial participation rate here for your review.
> > >
> > > Because GCFL requires careful parameter tuning and the difficulty in finding optimal results, combined with its performance on the Cora dataset being generally comparable to FedPer, we have not included it in this experiment. The results demonstrate that FlatLand maintains excellent performance even with reduced participation rates, highlighting the superiority of our approach.
> > >
> > > | Participate Rate |  0.1  |  0.3  |  0.5  |  0.7  | **1.0** |
> > > | :--------------: | :---: | :---: | :---: | :---: | :-----: |
> > > |      FedAvg      | 18.14 | 36.64 | 34.30 | 33.61 |  30.20  |
> > > | FedPer       | 56.03 | 60.22 | 74.79 | 73.48 | 68.11 |
> > > | FedHGCN      | 54.38 | 77.08 | 69.87 | 73.35 | 62.52 |
> > > |   **FlatLand**   | **81.82** | **81.11** | **79.27** | **79.42** |  **77.98**  |
> > >
> > > &nbsp;
> > >
> > > Best Regards,
> > >
> > > The Authors

---

> ### Author Response · Authors · 2024-11-22
> **Response (3/4)**
>
> Thank you for your questions. Next, we will summarize the questions with "**Q:**" and our responses with "**R**:"
>
> &nbsp;
>
> ---
>
>
> **Q1:** Why do the $v$ parameters not appear anywhere in Equation (4)?
>
> **R:** The parameter $v$ is introduced to enable a more generalized formulation, supporting advanced non-linear operations with learnable parameters. While dropout is presented as a simple example without $v$ , the parameter can play a role in other minor operations within hyperbolic space. Specifically, $v$ can function as learnable parameters in normalization operations, constraining the norm via $ \sigma(v^T x) $ to prevent excessive growth of hyperbolic embeddings, or as bias terms added to $ x $. In later sections, the presentation is simplified to focus on the core linear transformation aspects (lines 283–284), as these additional operations do not influence the fundamental principles of our method. We will add this explanation in the latter version.
>
> &nbsp;
>
> ---
>
> **Q2:** How does Lorentz space actually model heterogeneity? Provide theoretical or empirical evidence.
>
> **R: From a theoretical perspective.** There is a direct correlation between graph distributions and the curvature of hyperbolic geometry [1]. Specifically, the stronger the power-law distribution of graph degrees, the more the data deviates from Euclidean space, corresponding to a larger curvature of Hyperbolic space. This makes graph data particularly well-suited for validating our method's ability to address complex geometric properties, supported by strong theoretical motivation and guarantees. For data in other domains, there is also a series of works to empirically show the relationship between the hyperbolic curvature and the distribution [6-7].
>
> The theoretical guarantee of our decoupling strategy stems from the working principles of the Lorentz neural network. As shown in Equation (4), only the time-like dimension of the representation ($x_t$) is directly influenced by the curvature $-K$. Since curvature reflects the overall distribution of the data, it is straightforward and reasonable to assume that $x_t$ captures the heterogeneity of the data.
>
> Moreover, in Section 6 (*Perspective on Lorentz transformations*), we provide a relativistic perspective on the distinction between $x_t$ and $x_s$. Specifically, Lorentz space with different curvature represents varying spacetime intervals, and changes in the same event across different intervals primarily manifest in the time-like dimension. This further supports our hypothesis that $x_t$ effectively encodes data heterogeneity.
>
> **From an empirical perspective.** We conducted a series of experiments to validate the ability of our method to decouple data heterogeneity. First, we aggregated personalized data during training and observed significant performance fluctuations, with results deteriorating as shown in Figure 4. Second, we applied the same approach in Euclidean space, but it failed to achieve significant improvements. These observations help validate the effectiveness of our method in leveraging Lorentz space to handle heterogeneous data.
>
> &nbsp;
>
> ---
> **Q3:** Why do you decouple the parameters the way you do?
>
> **R:** Thank you for your question. Our decoupling strategy focuses on separating the parameters at each layer of the model. As discussed in our response to **Q2**, the key component that captures data heterogeneity is **the time-like dimension ($x_t$) of the hyperbolic embedding.** After each layer of the Lorentz neural network, the input vectors are transformed into a new hyperbolic embedding, with the time-like dimension updated as $x_t^{(l+1)} = \sqrt{||mx_t^{(l)} + \mathbf{M}\mathbf{x}_s^{(l)}||^2+K}$. This means that the component at layer $(l+1)$ that carries the heterogeneity information is transferred as $x_t^{(l+1)}$. Thus, the parameters associated with $x_t^{(l+1)}$ are naturally aligned with the heterogeneity information. This ensures that our decoupling strategy is consistent with the theoretical derivation and can be effectively applied across multiple layers.
>
> &nbsp;
>
> ---
>
> **References:**
>
> [1] Krioukov, Dmitri, et al. "Hyperbolic geometry of complex networks." *Physical Review E—Statistical, Nonlinear, and Soft Matter Physics* 82.3 (2010): 036106.
>
> [6] Curvature Generation in Curved Spaces for Few-Shot Learning. Gao et al. (2021). ICCV
>
> [7] Curvature-Adaptive Meta-Learning for Fast Adaptation to Manifold Data. Gao et al. (2023). TPAMI

---

> ### Author Response · Authors · 2024-11-22
> **Response (4/4)**
>
> **Q4: Does the method work for larger numbers of clients, without full client participation each round?**
>
> **R:** Thank you for the good question. We have conducted experiments on the image dataset to evaluate the results of these scenarios in the response of **C2**.
>
> &nbsp;
>
> ---
>
>
> **Q5: Statistically significant?**
>
> **R:** Our method's significance is validated using p-values. Additionally, the standard deviation values are shifted to two decimal places to save space. We will provide clarification on this in the revised manuscript for better understanding.
>
> &nbsp;
>
> ---
>
>
> **Q6: Embedding dimension?**
>
> R: Thank you for pointing out. Yes, the embedding dimension is the hidden dimension of the model. We will further refine our writing and try to provide as much clear explanation as possible within the limited space.

---

> ### Author Response · Authors · 2024-11-22
> **Summay**
>
> Our work introduces **a novel geometric perspective in personalized federated learning, particularly for addressing data heterogeneity**. As the first work exploring this direction, we provide valuable insights for the field, as acknowledged by multiple reviewers.
>
> - Reviewer 4tQH noted that "the idea in this paper is interesting and inspiring,"
>
> - Reviewer Wtgh highlighted "the novelty of increasing dimensionality to embed client data in Lorentz space,"
>
> - Reviewer Dc6W recognized our work as "an innovative approach to personalized federated learning by leveraging hyperbolic geometry, specifically through the use of Lorentz spaces with tailored curvatures.”
>
> &nbsp;
>
> Overall, the proposed decoupling strategy has been demonstrated to effectively mitigate data heterogeneity challenges and enhance the performance of clients’ local Lorentz neural network models **with strong motivation and theoretical support**. And our approach **achieves superior performance**, particularly on tree-like structures and power-law distributed data such as graphs, where the inherent hierarchical nature aligns well with hyperbolic geometry. We believe our analyses and ablation experimental results strongly validate the effectiveness of our proposed approach.
>
> &nbsp;
>
> Though our method can be readily transferred to other fields that use linear neural networks, we want to clarify that **our primary goal is not to achieve SOTA performance across all data and tasks**, as it also depends on the performance of the local hyperbolic backbone. Hyperbolic space may not be universally optimal for all underlying data distributions, as its effectiveness depends on whether the data can be appropriately modeled by varying negative curvature. For instance, certain datasets may exhibit inherent positive curvature characteristics [8-9]. This observation suggests a **promising future research direction**: modeling more complex data structures in mixed-curvature spaces. We will explore this method in the future, as claimed in our conclusion section.
>
> &nbsp;
>
> ---
> **References:**
>
> [8] Learning mixed-curvature representations in product spaces. ICLR. 2019
>
> [9] Pseudo-Riemannian Graph Convolutional Networks. NeurIPS 2022

---

> ### Author Response · Authors · 2024-11-25
>
> Thank you very much for your following-up questions.
>
> &nbsp;
>
> **Metric used in Participation Rate**: For the graph dataset, we are using classification accuracy, which is consistent with the main experiment table. For MNIST, we are using Accuracy (Acc) as the metric.
>
> &nbsp;
>
>
> **How AUC is calculated?** We directly followed the computation method used in the PFLib library [1]. After examining the source code, the specific implementation is:
>
> ```
> auc = sklearn.metrics.roc_auc_score(y_true, y_prob, average='micro')
> ```
>
> &nbsp;
>
> **About the baselines**:
> Due to time constraints, some baselines require hyperparameter tuning, and we are still conducting experiments. We will continuously update and include them in the final version. However, based on our presented experiments, we can observe that lowering the participation rate does not significantly impact our performance, which indicates that our method still works and performs well in this setting. Therefore, we believe this would NOT be a major concern for this work.
>
> &nbsp;
>
> Thank you again for all your valuable suggestions and questions. Please do not hesitate to let us know if there are any other questions and concerns.
>
> &nbsp;
>
> ---
> References:
>
> [1] PFLlib: Personalized Federated Learning Algorithm Library.

---

> > ### Author Response · Authors · 2024-12-02
> >
> > Dear Reviewer KA7L,
> >
> > &nbsp;
> >
> > As the review period is coming to a close, we wanted to follow up regarding our rebuttals. We believe we have thoroughly addressed your insightful comments and concerns through our responses. Furthermore, we will continue to enhance and refine our work for the final version.
> >
> > **Since ALL other reviewers hold quite positive feedback and have affirmed our novelty and contribution, your feedback is invaluable to us**. We kindly request that you **respect the contribution and dedication behind this work** and **consider raising the score** if there is no new concern. Please let us know if you need any additional information or clarification to complete your review.
> >
> > Thank you very much for your time and careful consideration of our work!
> >
> > &nbsp;
> >
> > Best regards,
> >
> > The Authors

---

> > > ### Comment · Reviewer_KA7L · 2024-12-02
> > > **Summary of Rebuttal Period**
> > >
> > > I would like to thank the authors for their answers to my questions as well as for providing additional work and experiments over the course of the rebuttal period. Unfortunately, I am unable to increase my score and cannot recommend that the paper be accepted due to what I believe to be major outstanding problems with the current submission. I expand on these in more detail below, as a summary they are:
> > >
> > > - Lack of theoretical contributions that show the method performs well or captures heterogeneity as claimed.
> > > - Lack of a broad enough experimental evaluation (I remain unconvinced that the method performs well in any other setting than graph datasets with small numbers of clients).
> > > - Proposed parameter decoupling does not achieve the separation of heterogeneous and homogeneous parts of the data as the authors claim.
> > >
> > > **Theory**
> > >
> > > I am still unconvinced that there is any meaningful theoretical contribution. The authors explanation for why Corollary 1 is necessary does not really make sense to me: LT(x, M) is designed by prior work so that the output always lies in Lorentz space (for any M). It doesn’t matter that M is obtained by federated averaging. The authors themselves note that they do not claim the theory to be a major contribution.
> > >
> > > **Experiments**
> > >
> > > There are two major outstanding issues for me within the experimental evaluation:
> > >
> > > 1. I believe that as presented the method works poorly in cross-device federated learning.
> > >
> > > Setting aside the fact that the method requires clients to maintain internal states throughout the optimization (which is already undesirable for cross device FL), the most critical reason why the method is poorly suited to cross-device FL is that I believe it is badly impacted by partial client participation. This is because the personalized portions of the parameters are not updated if a client does not participate. So for example if a client that has not yet participated in training, joins at a later stage (something that is perfectly normal to happen in cross-device FL), they will start with randomly initialized parameters in many parts of their network, while other clients may hold near optimal parameters. This is likely to cause issues with highly heterogeneous updates.
> > >
> > > The authors were unable to provide convincing evidence that this does not occur. In fact the evidence that they provided during the rebuttal made me more convinced this is an issue. For instance in the partial client participation table for MNIST their method drops about 5.5% in accuracy moving to 0.1 participation, while FedAvg drops less than 1%. The authors did not provide the results on additional baselines, I find it likely that they perform much better than Flatland in this setting. Also I would like to emphasize that 100 clients on MNIST is a very easy FL setting. This drop will be much more severe on more challenging federated datasets with much larger numbers of clients.
> > >
> > > Contrary to what the authors claim, being able to handle partial client participation is absolutely crucial if your method is to work in a cross device setting (potentially millions of devices with very low participation rates).
> > >
> > > 2. I do not see any convincing evidence that the method works well on data modalities other than graph datasets.
> > >
> > > I thank the authors for providing additional experiments on MNIST but I feel this is not enough to support the claim that the method works well on non-graphical data. Put simply, MNIST with 100 clients is not a challenging or realistic enough FL setting to claim the method performs well on image data, which is part of what the authors are claiming.
> > >
> > > Combining these two issues together, the paper is only able to demonstrate empirical performance in a cross-silo (few clients, full client participation) graph dataset setting which, given the lack of theoretical contribution, is not enough in my opinion.
> > >
> > > **Method**
> > >
> > > The primary methodological contribution of the method is the parameter decoupling scheme. However, the reason for this decoupling as stated by the authors still doesn’t make sense to me. As shown in equation (4), the space-like dimensions of the output of the first layer are dependent on the heterogeneous portion of the input $x_t$. These will then be acted on by the shared parameters of the network from layer 2 onward. Likewise, the time dimension of the output is also dependent on the space dimension of the input. So in a network with more than a single layer, the proposed parameter decoupling scheme does not do what the authors state it is intended to do: namely that the personalized parameters handle the heterogeneity ($x_t$) and the shared parameters handle the homogeneous part of the data.

---

> ### Author Response · Authors · 2024-12-01
>
> Dear Reviewer KA7L,
>
> &nbsp;
>
> We sincerely thank you for your efforts in helping us improve our paper! With only two days remaining, we are kindly reaching out to request your further feedback. **We believe we have thoroughly addressed all your concerns and respectfully hope you consider raising your score.**
>
> Thank you once again for your efforts in ensuring the high standards of academic excellence. **We look forward to your response and remain available for any further discussions.**
>
> &nbsp;
>
> Best Regards,
>
> The Authors

---

> ### Author Response · Authors · 2024-12-03
> **Follow-up Rebuttals (1/2)**
>
> Dear Reviewer KA7L,
>
> We appreciate your time and effort in reviewing our manuscript. However, we respectfully disagree with your follow-up evaluation and would like to address some misconceptions.
>
> &nbsp;
>
>
> **1. Theoretical Contribution.**
>
> First, regarding the theoretical contribution, specifically Corollary 1: We **NEVER** claimed this as our primary contribution even in the original manuscript. Rather, it serves as a necessary foundation in hyperbolic networks, which is not as trivial as suggested in your review. A thorough understanding of hyperbolic geometry would reveal its significance. The series of analyses we provided, including but not limited to Corollary 1, contributes to the completeness and correctness of our entire work. We also add a convergence analysis accordingly in the revised version. These theoretical foundations are essential components that support our main contributions and methodological innovations.
>
> And what is important is that our method is designed and evaluated supported by strong theoretical motivation, as we explained in responses 1/4 and 3/4.
>
> &nbsp;
>
> **2.1 Experimental Settings.**
>
> Regarding experimental validation, we want to clarify that we **NEVER** claimed our method works universally across **ALL** datasets. We explicitly acknowledged limitations and future directions in both our original submission and revised version.  And explanations are also provided in the rebuttal (Response 2/4 The reason we focus on graph data) and we also added this clarification in the Future Work Section that *"Note that hyperbolic space is not universally optimal for all data distributions — some exhibit positive curvature — highlighting the need to model complex data structures in mixed-curvature spaces"*.
>
> &nbsp;
>
> Regarding the experimental settings with up to 50 clients: We **NEVER** claim that work in a **cross-device setting** is our main contribution. Instead, we follow the graph federated learning settings to solve personalized FL problems, which are seldom considered in this scenario. This should not be considered a major concern, as **it aligns with current standards in federated graph learning research, as demonstrated in accepted papers \[1-2\]**. More importantly, our comprehensive experimental results demonstrate excellent performance across these settings even in low partial participation rates in graph dataset, which strongly validates our hypotheses and the effectiveness of our proposed method.
>
> &nbsp;
>
> **2.2 Partial participation scenarios.**
>
>
> We **respectfully strongly disagree with your assertion** that our method cannot work in partial participation scenarios for the following reasons:
>
> From the experimental results, our experimental results demonstrate strong and consistent performance, particularly in our emphasized settings, **yet these significant results seem to have been overlooked in your evaluation**. While you highlighted the MNIST results, we want to emphasize that the image domain dataset was not our primary focus, as we clearly stated in our response section. Due to time constraints, we haven't exhaustively explored all possible parameter configurations to find the optimal settings in image datasets (the results have been updated). We will explore more possibilities in future work.
>
> Moreover, the **key issue** you mentioned is the scenarios where *"the personalized portions of the parameters for the later joined clients are not updated."* This problem is **not the focus of the research on personalized federated learning (PFL)**. In fact, the extreme situation you mentioned **occurs for all parameter decoupling methods, including FedPer, which is one of the most widely used and popular methods in PFL**. This is a separate research problem, and our method is orthogonal to it. We believe relevant strategies for handling this issue are not in conflict with our method. Our decoupling strategy, similar to FedPer, effectively handles this issue by separating heterogeneous and homogeneous information but with enhanced interpretability and better heterogeneity handling.
>
> Nevertheless, we are willing to provide **a solution of how our method can solve this issue easily**. **When a new client joins**, we can directly use the global shared parameters as the initialized shared parameters. Compared to FedPer, an advantage of our method is that **we can pre-estimate the curvature of the client's data and directly fetch the well-trained personalized parameters of other clients with similar data curvature and use them as initialization**. This not only avoids parameter optimization performance issues but also ensures privacy. In contrast, FedPer cannot effectively tell which personalized parameters are better or more useful for new clients. **This, in turn, highlights a potential advantage of our method** in this scenario compared to ordinary parameter decoupling methods, as we utilize geometry information, further underscoring the inspiring nature of our approach.

---

> ### Author Response · Authors · 2024-12-03
> **Follow-up Rebuttals (2/2)**
>
> &nbsp;
>
> **3. Decoupling strategy.**
>
> **The heterogeneity information is not simply captured by the input $x_t$, but rather by the time dimension representation in the hyperbolic space during the Lorentz Transformation process**. The key component that captures data heterogeneity is **the time-like dimension of the hyperbolic embedding** rather than the original input $x_t$. The time-like component of the output representation is calculated using the norm of the space-like dimensions and the curvature, as shown in Equation (4) of our manuscript. Its special computation method determines its uniqueness on the hyperbolic manifold. Therefore, after multiple layers of output, the 0-th dimension of the embeddings truly captures the heterogeneity information we hypothesize, and this is entirely consistent with our derivation.  We have provided a detailed explanation of this in the rebuttal (**Response (3/4) Q2 and Q3**).  **We are confident** that this is an intuitive and easily understandable concept within the field of hyperbolic neural networks.
>
> Most importantly, our **extensive experimental results, including thorough ablation studies, validate the significance and effectiveness of our proposed decoupling strategy**. These empirical results provide concrete evidence supporting the merits and performance of our method, which we believe deserves proper acknowledgment and recognition. **This is also acknowledged by your previous comments:** "*The authors provide ablation studies examining the impact of different components of the method*". Additionally, this is also mentioned by
> * Reviewer nTmN: *Experimental study of the paper looks convincing*
> * Reviewer Wtgh: *Experiments are extensive*
>
> &nbsp;
>
> ---
>
>
> In summary, in our opinions, all the issues you mentioned above **are not the core contributions of our paper**, and **some are not even core research problems in the field of personalized federated learning** (such as late-joining clients and partial participation). This does not affect the integrity and contribution of our work. We would like to re-emphasize that :
>
> 1. The core contribution of our work is being the first to leverage geometric information for decoupling personalized information, providing new insights into this field, as highlighted by
> 	* Reviewer 4tQH: "I think this work can inspire more ideas in federated hyperbolic learning problems."
> 	* Reviewer Dc6W: "I still find this approach offers valuable insights, particularly in new domains, and I will maintain my score."
>
> 2. Our choice to conduct experiments on graphs is well-supported by theoretical and intuitive justifications, better validating our hypotheses (Response 2/4 1). Furthermore, our **experimental scopes are consistent with those of previous methods**. Our work is coherent and complete within the scope of our claimed contributions.
>
> 3. We acknowledged that hyperbolic spaces might not be effective for ALL data types (which is not our contribution or focus). We have provided sufficient reasoning and potential future research directions regarding this. "*Note that hyperbolic space is not universally optimal for all data distributions — some exhibit positive curvature — highlighting the need to model complex data structures in mixed-curvature spaces.*"
>
> 4. Partial training participation is not a key point in the point of personalized federated learning. We have already thoroughly verified the feasibility of our method for handling partial training participation in graph dataset including all strong baselines. For extreme scenarios, we believe our method still has great potential, as we explained before, which could be explored in future work, although it is not a core focus of the personalized federated learning field.
>
> Thank you again for all your comments. We would greatly appreciate a reconsideration based on these clarifications and the actual merits of our work.
>
> &nbsp;
>
> Best regards,
>
> The Authors
>
> &nbsp;
>
> ---
>
> **References:**
>
> [1] Subgraph Federated Learning with Missing Neighbor Generation
>
> [2] An efficient federated learning framework for graph learning in hyperbolic space

---

### Official Review · Reviewer_Dc6W · 2024-11-03

**Soundness:** 3
**Presentation:** 2
**Contribution:** 3
**Rating:** 6
**Confidence:** 4

**Summary:**

This paper introduces FlatLand, a novel personalized federated learning (PFL) approach that addresses data heterogeneity across clients by leveraging hyperbolic geometry, specifically Lorentz space. Unlike traditional PFL methods that operate in Euclidean space, FlatLand embeds each client's data in a tailored Lorentz space with different curvatures to better capture non-Euclidean properties of the data. Instead of explicitly calculating client similarities or using additional modules for personalization, FlatLand leverages the geometric properties of Lorentz space to handle heterogeneity naturally through different curvatures. Also, they use parameter decoupling for shared and local models, similar to some other personalized FL models.

**Strengths:**

This paper presents an innovative approach to personalized federated learning by leveraging hyperbolic geometry, specifically through the use of Lorentz spaces with tailored curvatures for different clients. The theoretical foundation is well-developed, with mathematical proofs and insights connecting geometric properties to data heterogeneity. The experimental results show promising improvements over existing methods, particularly in low-dimensional settings, which could be valuable for resource-constrained environments. The parameter decoupling strategy based on geometric properties is elegant and eliminates the need for explicit client similarity calculations, which is a common computational burden in some of the other personalized federated learning approaches using clustering approaches.

**Weaknesses:**

The paper's evaluation primarily focuses on graph-based tasks, leaving questions about its generalizability to other types of data and models. While they mention the potential applicability to other domains, this claim remains unverified even in the experimental sense. The method introduces additional complexity through hyperbolic geometry and requires curvature estimation, which could make implementation and tuning more challenging in practice. The computational overhead of working in Lorentz space compared to Euclidean space is not thoroughly discussed, and the paper doesn't address potential numerical stability issues that often arise when working with hyperbolic spaces. Additionally, while the method shows improvements over baselines, some of the gains are modest, particularly in the graph classification tasks, suggesting that the added complexity might not always justify the performance benefits.
The main concerns of the paper are as follows:
 - The paper presents its parameter decoupling strategy as a novel contribution without comparing it with similar approaches in existing literature. The absence of comparisons with previous decoupling methods, particularly [A], makes it difficult to assess the true novelty and advantages of their approach.
- The paper lacks clarity in explaining crucial algorithmic details, particularly regarding curvature estimation. While curvature is central to the method's performance, the paper doesn't adequately explain how it's estimated, updated, or integrated into the training process. The absence of analysis on the impact of different curvature values and their stability leaves important implementation questions unanswered.
- The evaluation is heavily focused on graph-based tasks, with no experimental validation on other common federated learning applications like computer vision or natural language processing (such as FEMNIST). This narrow focus raises questions about the method's applicability and effectiveness in broader contexts, especially given the additional complexity introduced by hyperbolic geometry.
- The paper lacks a thorough analysis of computational overhead compared to Euclidean-based methods. There's no discussion of memory requirements for storing different Lorentz spaces or how the method scales with an increasing number of clients. The communication costs and practical implications in real-world deployments remain unexplored.


[A] Liam Collins, Hamed Hassani, Aryan Mokhtari, and Sanjay Shakkottai. Exploiting shared representations for personalized federated learning. In ICML, volume 139 of Proceedings of Machine
Learning Research, pp. 2089–2099. PMLR, 2021.

**Questions:**

In addition to the questions and concerns above, I have some other questions:

- Could you elaborate on how the curvature parameters are practically updated during the federated learning process? Specifically, what happens if the estimated curvature is unstable or changes significantly between rounds, and how do you ensure this doesn't negatively impact the model's convergence?
- Could you discuss any potential challenges or modifications needed when applying your method to these domains, particularly regarding the relationship between data heterogeneity and geometric properties in these different contexts?
- Could you provide insights into the scalability of your approach as the number of clients increases, especially considering the additional overhead of maintaining different Lorentz spaces? For instance, provide some results with training times and memory utilization.

---

> ### Author Response · Authors · 2024-11-21
> **Response (1/4)**
>
> Thank you for your valuable comments and suggestions. Below, we will summarize the concerns with "**C:**" and our responses with "**R:**"
>
> &nbsp;
>
> ---
>
> **C1:** Motivation about focusing on graph data and generalizability to other types of data and models.
>
> R: In this work, we specifically chose to focus on graph data because graphs inherently exhibit non-Euclidean structures and significant heterogeneity, as demonstrated in previous theoretical analyses. Besides, the heterogeneity is directly related to the curvature of hyperbolic geometry [1]. This makes graph data particularly suitable for validating our method's effectiveness in handling complex geometric properties **with a clear motivation and theoretical guarantee**.
>
> The main reason that we didn't compare with FedRep in the main paper is due to their lack of graph-related implementations. With the limited time, we have conducted additional experiments on MNIST to demonstrate our method's broader applicability. The results are as follows:
>
> | **Dataset** | **#Clients** | **FedAvg (\%)** | **FedProx (\%)** | **Ditto (\%)** | **GPFL (\%)** | **FedRep (\%)** | **FedCAC (\%)** | **FlatLand (\%)** |
> | --- | --- | --- | --- | --- | --- | --- | --- | --- |
> | MNIST (Acc) | 20 | $87.86 \pm 0.0816$ | $87.53 \pm 0.0771$ | $97.85 \pm 0.0191$ | $92.90 \pm 0.0724$ | $98.14 \pm 0.0196$ | $97.85 \pm 0.0189$ | $\mathbf{98.35} \pm 0.0136$ |
> | MNIST (AUC) | 20 | $97.77 \pm 0.0149$ | $98.81 \pm 0.0110$ | $99.92 \pm 0.0012$ | $99.48 \pm 0.0110$ | $99.85 \pm 0.0196$ | $99.92 \pm 0.0012$ | $\mathbf{99.93} \pm 0.0011$ |
> | MNIST (Acc) | 100 | $86.14 \pm 0.2066$ | $84.50 \pm 0.1658$ | $96.45 \pm 0.0415$ | $96.52 \pm 0.0462$ | $96.54 \pm 0.0750$ | $96.59 \pm 0.0505$ | $\mathbf{96.64} \pm 0.0495$ |
> | MNIST (AUC) | 100 | $96.57 \pm 0.0508$ | $98.22 \pm 0.0442$ | $99.78 \pm 0.0047$ | $99.70 \pm 0.0136$ | $99.67 \pm 0.0190$ | $\mathbf{99.81} \pm 0.0012$ | $99.70 \pm 0.0116$ |
>
> These results demonstrate that FlatLand performs competitively even on image data. We plan to explore FlatLand more for image and text datasets and analyze the detailed relation between the curvature and the specific data properties in future work.
>
> &nbsp;
>
> ---
>
> **Reference:**
>
> [1] Krioukov, Dmitri, et al. "Hyperbolic geometry of complex networks." *Physical Review E—Statistical, Nonlinear, and Soft Matter Physics* 82.3 (2010): 036106.
>
> [2] Bold but cautious: Unlocking the potential of personalized federated learning through cautiously aggressive collaboration, ICCV 2023
>
> [3] Gpfl: Simultaneously learning global and personalized feature information for personalized federated learning, ICCV 2023

---

> ### Author Response · Authors · 2024-11-21
> **Response (2/4)**
>
> **C2:** Computational efficiency. *“The method introduces additional complexity through hyperbolic geometry and requires curvature estimation, which could make implementation and tuning more challenging in practice. The computational overhead of working in Lorentz space compared to Euclidean space is not thoroughly discussed”*
>
> **R:** Let us clarify the complexity analysis of FlatLand compared to FedAvg:
>
> 1. **Local updates.** the additional operations in FlatLand (curvature estimation and exponential map). **The curvature estimation does not introduce significant computational overhead** since it only needs to be **computed once** for each client's data distribution. This value can be pre-computed and reused. Similarly, the exponential map transformation only requires a single non-linear mapping operation based on input sample norms with a complexity of $O(1)$, which can also be pre-computed and cached for efficiency.
>
> 2. **Aggregation**, FlatLand maintains the same time complexity as FedAvg when using equivalent hidden dimensions. While FlatLand introduces time-like space parameters, it only aggregates shared parameters while maintaining personalized parameters locally. Importantly, as shown in Section 7.3, FlatLand performs better in low dimensionality, potentially reducing communication costs in practice.
>
> 3. **Storage requirements**, FlatLand requires $O(d+1)$ additional storage per client compared to FedAvg, accounting for the time-like dimension and curvature parameter, where $d$ is the hidden dimension. This overhead is minimal since d is typically small, and FlatLand's superior performance in low-dimensional settings further mitigates practical storage concerns.
>
> Overall, while FlatLand adds a few steps compared to FedAvg, the computational overhead is limited due to pre-computation and efficient constant-time operations.  Consequently, this **does not affect the scalability of the method**. Moreover, our experiments show that its benefits outweigh the slight complexity increase, particularly for managing heterogeneous data in resource-constrained, low-dimensional settings. We have added this analysis in Appendix B.6.
>
> &nbsp;
>
> ---
>
> **C3:** Numerical stability. *“The paper doesn't address potential numerical stability issues that often arise when working with hyperbolic spaces.”*
>
> **R: Regarding numerical stability**, our method specifically employs fully Lorentz neural networks, which were designed to address the numerical instability issues commonly associated with hyperbolic neural networks. Traditional hyperbolic approaches often face instability due to frequent projections between tangent and hyperbolic spaces [5]. However, fully Lorentz neural networks perform operations directly in Lorentz space without any further mapping functions, significantly reducing these stability concerns.
>
> &nbsp;
>
> ---
>
> **C4:** Differences with previous decoupling methods, particularly FedRep
>
> **R:** We would like to further highlight our novelty. While FedRep does propose a parameter decoupling approach in Euclidean space, our method differs fundamentally in both motivation and implementation.
>
> **Our decoupling strategy is uniquely motivated by the geometric properties of Lorentz space**, where the time-like dimension naturally captures heterogeneous information. This geometric interpretation, as illustrated in Figure 1(b), allows us to separate client-specific features (manifested in the time-like dimension) from shared information (in space-like dimensions) without requiring explicit similarity calculations or additional modules.
>
> In contrast, FedRep focuses on finding shared representations in Euclidean space through optimization constraints. While both methods aim to separate shared and personalized components, **our approach leverages the inherent structure of hyperbolic geometry to achieve this separation more naturally**. This is evidenced by our theoretical analysis in Section 6, which shows how the time-like parameter inherently captures client-specific information through the gradient calculations (Equations 11-13).
>
> Furthermore, our experimental results demonstrate the effectiveness of this geometry-driven approach, **particularly in scenarios with strong non-Euclidean properties**, as shown in the curvature analysis (Figure 6). We also conduct experiments that compare our method with FedRep in MNIST in the response to **C1**.
>
> &nbsp;
>
> ---
> **References:**
>
> [5] Hyperbolic graph convolutional neural networks. Chami, I., et al. (2019). NeurIPS

---

> ### Author Response · Authors · 2024-11-21
> **Response (3/4)**
>
> Thank you for your questions. Below, we will summarize the questions with “**Q:**” and our responses with "**R:**"
>
> &nbsp;
>
> ---
>
>
> **Q1:** Curvature estimation implementation.
>
> R:  For curvature estimation, we employ Forman-Ricci curvature as described in Section 5.1. The motivation behind this choice is that Forman-Ricci curvature effectively captures how significantly a graph's structure deviates from Euclidean geometry, and has a theoretical connection with the curvature of hyperbolic geometry [1]. As shown in Figure 6, different clients exhibit varying curvature values, reflecting their distinct structural properties.
> For practical implementation, we treat curvature as a learnable parameter initialized using the Forman-Ricci estimate. Specifically, we use σ(K) + 0.5 to ensure the curvature remains negative and within a well-performing range of [0.5, 1.5]. This range has been empirically validated in previous hyperbolic learning literature and helps maintain numerical stability while accommodating heterogeneous data distributions [4-7].
>
> The effectiveness of this approach is demonstrated in our ablation study (Figure 4), where comparing against a fixed curvature setting ("w/o TS") shows that adaptive curvature significantly improves performance and does not lead to heavy fluctuation, which further shows the stability of this method.
>
> Moreover, we theoretically analyze the convergence rate of FlatLand and demonstrate that the parameter $K$ has almost no impact on the convergence rate. Detailed analyses are added in Appendix B.5.
>
> &nbsp;
>
> ---
>
> **References:**
>
> [4]  Fully hyperbolic neural networks. Chen, W., et al. (2021). ACL
>
> [5] Hyperbolic graph convolutional neural networks. Chami, I., et al. (2019). NeurIPS
>
> [6] HGCF: Hyperbolic graph convolution networks for collaborative filtering. Sun, J., et al. (2021). WWW
>
> [7]  Discrete-time temporal network embedding via implicit hierarchical learning in hyperbolic space. Yang, M., et al. (2021). KDD

---

> ### Author Response · Authors · 2024-11-21
> **Response (4/4)**
>
> **Q2:** Potential challenges or modifications needed when applying your method to other domains.
>
> **R:** Different data modalities exhibit varying levels of non-Euclidean characteristics. While graph data naturally possesses strong non-Euclidean properties (as shown by our Ricci curvature analysis in Figure 6), other domains like images and text also demonstrate heterogeneous structures. For instance:
>
> 1. In vision tasks, the manifold of natural images often exhibits hierarchical relationships (e.g., different scales of visual features) that can benefit from hyperbolic representation. The heterogeneity often comes from varying visual styles, lighting conditions, or camera angles.
> 2. In text data, semantic hierarchies and power-law distributions in word frequencies naturally align with hyperbolic geometry's ability to embed tree-like structures. The heterogeneity might arise from different writing styles, topics, or languages.
>
> While our work represents a promising start in leveraging geometric information for personalized federated learning, we acknowledge that hyperbolic space might not be optimal for all types of data. Analyzing the scenarios and types of heterogeneity where hyperbolic geometry is most advantageous remains a promising direction for future work. Besides, we can explore spaces with mixed curvature (including positive and negative curvatures simultaneously) to better accommodate different data geometries. $\mathsf{FlatLand}$'s fundamental approach to addressing heterogeneity through geometric modeling provides a solid foundation for these extensions.
>
> &nbsp;
>
> ---
>
> **Q3:** Scalability considering the additional overhead.
>
> **R:** We provide a detailed analysis of the time complexity cost and memory usage of our method compared to the simplest FedAvg method, as addressed in our response to C2. This analysis highlights the scalability of our approach. Additionally, we conducted experiments on data from 100 clients, further validating the extensibility of our method.
>
> &nbsp;
>
> ---
>
> We hope we've addressed your key concerns, particularly regarding the generalizability of our method beyond graph data, the complexity of curvature estimation, and computational efficiency. We demonstrated that FlatLand performs well on benchmarks like MNIST, with minimal computational overhead due to pre-computation (Appendix B.6.). Additionally, we clarified that curvature estimation is stable and does not impact convergence (Appendix C.2. and Appendix B.5.).

---

> ### Author Response · Authors · 2024-12-01
>
> Dear Reviewer Dc6W,
>
> &nbsp;
>
> We sincerely thank you for your positive feedback and efforts in helping us improve our paper! With only two days remaining before the deadline and having not yet received a response to our carefully formulated point-to-point replies, we kindly request your feedback. If you have any further questions or suggestions, we welcome the opportunity to discuss and address them.
>
> **We believe we have thoroughly addressed all the concerns raised in your initial review. And we would be grateful if you would consider raising your score accordingly.**
>
> Thank you once again for your efforts in ensuring the highest standards of academic excellence. **We look forward to your response and remain available for any further discussions.**
>
> &nbsp;
>
> Best Regards,
>
> The Authors

---

> > ### Comment · Reviewer_Dc6W · 2024-12-03
> > **Reviewing feedback**
> >
> > Thank you for the detailed explanation. I’ve reviewed your points that addressed most of my concerns. However, I remain somewhat skeptical about the comparison between your method and FedRep. While I appreciate your results on the MNIST dataset, I believe this dataset is not entirely representative of the broader applicability of different methods, and a more thorough analysis would strengthen your claims. That said, I still find this approach offers valuable insights, particularly in new domains, and I will maintain my score.

---

> > > ### Author Response · Authors · 2024-12-03
> > > **Thank you!**
> > >
> > > Dear Reviewer Dc6W,
> > >
> > > &nbsp;
> > >
> > > Thank you very much for your valuable feedback and suggestions.
> > >
> > > Due to time constraints, we acknowledge that experimenting on the MNIST dataset alone is not sufficient to fully validate the method in other domain datasets. These were the basic initial results. We will explore more and conduct more experiments further, and we have emphasized this point in the revised manuscript: "*Note that hyperbolic space is not universally optimal for all data distributions — some exhibit positive curvature — highlighting the need to model complex data structures in mixed-curvature spaces.*"
> > >
> > > The core of this method was validated on graph datasets, demonstrating its effectiveness. In the future, we will explore its performance on more domain data and scenarios and design improved methods accordingly. As the development of hyperbolic spaces in image, text, and multimodal applications, we also believe that our idea could offer valuable insights for further design of personalized federated learning algorithms.
> > >
> > > &nbsp;
> > >
> > > Best Regards,
> > >
> > > The authors

---

### Official Review · Reviewer_Wtgh · 2024-11-05

**Soundness:** 2
**Presentation:** 3
**Contribution:** 3
**Rating:** 6
**Confidence:** 3

**Summary:**

This paper proposes a new personalized federated learning approach called FlatLand. Unlike current methods that commonly operate in Euclidean space, FlatLand embeds client data in a Tailored Lorentz space to more effectively represent the clients’ data distributions. Building on this foundation, the authors introduce a parameter decoupling strategy to aggregate shared parameters efficiently. Extensive experiments demonstrate the effectiveness of FlatLand.

**Strengths:**

1. The paper is well-written and easy to follow.
2. The idea of increasing dimensionality to embed client data in Lorentz space is novel.
3. Experiments are extensive.

**Weaknesses:**

1. Techniques that personalize client models by creating localized models or layers are highly relevant to this paper. However, the related work section includes somewhat outdated references. It would benefit from incorporating more recent approaches, such as FedCAC [1] and GPFL [2]. Additionally, the experiments lack comparisons with these types of methods.

2. This paper primarily focuses on graph datasets. Can FlatLand effectively perform on commonly used benchmarks in PFL, such as CV datasets (e.g., CIFAR-100) and text datasets (e.g., AGNEWS)?

[1] Bold but cautious: Unlocking the potential of personalized federated learning through cautiously aggressive collaboration, ICCV 2023
[2] Gpfl: Simultaneously learning global and personalized feature information for personalized federated learning, ICCV 2023

**Questions:**

Please see the weaknesses above.

---

> ### Author Response · Authors · 2024-11-21
> **Response**
>
> Thank you for your valuable comments and suggestions. Below, we will summarize the weaknesses with "**W:**" and our responses with "**R:**"
>
> &nbsp;
>
> ---
>
> **W1:** Baselines selection.
>
> **R:** Thank you for this suggestion. Note that our primary goal was to highlight and evaluate the benefits of integrating hyperbolic geometry in federated learning. Hence our comparisons focused on baselines with strong performance on graph data to ensure a fair evaluation. FedCAC and GPFL perform well in image datasets but lack graph-related implementations. Therefore, we compare our method with FedCAC and GPFL in image dataset; please check the response to W2.
>
> We will incorporate these clarifications into the revised manuscript to provide a clearer context for our focus and methodology.
>
> &nbsp;
>
> ---
>
> **W2:** This paper primarily focuses on graph datasets. Can FlatLand effectively perform on commonly used benchmarks in PFL?
>
> **R:** Thank you for your suggestions. We first would like to demonstrate the reason we chose graph data to conduct experiments and analysis.
>
> **1. The reason for choosing graph data.**
>
> In this work, we chose to focus on graphs because *graphs inherently exhibit non-Euclidean structures and significant heterogeneity, which are directly related to curvature*. This makes graph data particularly suitable for validating the effectiveness of our hyperbolic geometry-based approach in addressing data heterogeneity.
>
> * Theoretical foundation: The use of hyperbolic spaces to model varying graph structures is well-supported by theory. As demonstrated in [1], the relationship between graph structure and hyperbolic space provides a strong mathematical basis for our method's effectiveness.
>
> *  Clear demonstration of benefits: Graph data is ideal for showing our method’s ability as they generally have different curvatures. It also allows us to demonstrate the effectiveness of our parameter decoupling strategy in a setting where heterogeneity is more evident.
>
> **2. Applicability to other domains.**
>
> Our method can be easily applied to other data as our method directly adds one extra dimension to the linear layer. With limited rebuttal time, we have conducted additional experiments on MNIST to primarily demonstrate our method's broader applicability. The results are promising:
>
> | **Dataset** | **#Clients** | **FedAvg (\%)** | **FedProx (\%)** | **Ditto (\%)** | **GPFL (\%)** | **FedRep (\%)** | **FedCAC (\%)** | **FlatLand (\%)** |
> | --- | --- | --- | --- | --- | --- | --- | --- | --- |
> | MNIST (Acc) | 20 | $87.86 \pm 0.0816$ | $87.53 \pm 0.0771$ | $97.85 \pm 0.0191$ | $92.90 \pm 0.0724$ | $98.14 \pm 0.0196$ | $97.85 \pm 0.0189$ | $\mathbf{98.35} \pm 0.0136$ |
> | MNIST (AUC) | 20 | $97.77 \pm 0.0149$ | $98.81 \pm 0.0110$ | $99.92 \pm 0.0012$ | $99.48 \pm 0.0110$ | $99.85 \pm 0.0196$ | $99.92 \pm 0.0012$ | $\mathbf{99.93} \pm 0.0011$ |
> | MNIST (Acc) | 100 | $86.14 \pm 0.2066$ | $84.50 \pm 0.1658$ | $96.45 \pm 0.0415$ | $96.52 \pm 0.0462$ | $96.54 \pm 0.0750$ | $96.59 \pm 0.0505$ | $\mathbf{96.64} \pm 0.0495$ |
> | MNIST (AUC) | 100 | $96.57 \pm 0.0508$ | $98.22 \pm 0.0442$ | $99.78 \pm 0.0047$ | $99.70 \pm 0.0136$ | $99.67 \pm 0.0190$ | $\mathbf{99.81} \pm 0.0012$ | $99.70 \pm 0.0116$ |
>
> We plan to conduct more experiments and analysis on more domain and tasks. This is also mentioned in the "Future Work" section of our manuscript.
>
> &nbsp;
>
> ---
>
> **Reference:**
>
> [1] Krioukov, Dmitri, et al. "Hyperbolic geometry of complex networks." *Physical Review E—Statistical, Nonlinear, and Soft Matter Physics* 82.3 (2010): 036106.

---

> ### Author Response · Authors · 2024-12-01
>
> Dear Reviewer Wtgh,
>
> &nbsp;
>
> We sincerely thank you for your positive feedback and efforts in helping us improve our paper! With only two days remaining before the deadline and having not yet received a response to our carefully formulated point-to-point replies, we kindly request your feedback. If you have any further questions or suggestions, we welcome the opportunity to discuss and address them.
>
> **We believe we have thoroughly addressed all the concerns raised in your initial review. And we would be grateful if you would consider raising your score accordingly.**
>
> Thank you once again for your efforts in ensuring the highest standards of academic excellence. **We look forward to your response and remain available for any further discussions.**
>
> &nbsp;
>
> Best Regards,
>
> The Authors

---

### Official Review · Reviewer_4tQH · 2024-11-07

**Soundness:** 4
**Presentation:** 2
**Contribution:** 3
**Rating:** 6
**Confidence:** 5

**Summary:**

This paper studies the federated graph learning problem, and proposes to model data heterogeneity across different clients as different Lorentz spaces with different curvatures. The authors adapt the Lorentz network into the FL framework, and propose to maintain dedicated trainable client-specific parameters to learn different client curvatures, which makes the new method FlatLand different from previous baselines. Simulation results and ablation studies are also presented in the paper to support the better performance of the new method.

**Strengths:**

Although there are already many works on federated hyperbolic learning, I found the idea in this paper still interesting and inspiring. The proposed method is concise yet intuitive, and the simulation results seem to suggest that it also works well in practice for graph-related problems.

**Weaknesses:**

The presentation of this paper can be improved.

- Firstly, some of the notations are not consistent in the paper and can make readers confusing. For example, line 286-290, if $x^{(l)}$ is in $\mathcal{L}^n_K$, then $x_s^{(l)}\in\mathbb{R}^n$ instead of $\mathbb{R}^d$; $m^{(l)}$ should be in $\mathbb{R}^{m}$ and $M^{(l)}$ should be in $\mathbb{R}^{m\times n}$. This type of inconsistency exists throughout the paper.

- Secondly, some of the statements in the paper are left unexplained, which makes me very confused. For example, line 167-168 "For instance, for dropout, the operation function is f(Wx, v) = W dropout (x).", where is the usage of the bias v? Another example is Equation (4), it seems that the first row of $\hat{M}^{(l)}$ is not used at all in each LT layer. Is something missing here? I assume $K$ is a separate learnable scalar to represent the curvature?

- Thirdly, how the authors use the proposed method to solve federated **graph** learning problem is not clearly explained in the main text, as the proposed method is suitable for transforming the node feature, but is not easy to handle the adjacency matrix directly. So it would be confusing if the author just apply the method over graph datasets as presented in the simulation section. In appendix, I find that "For the node classification task, we employ 2-layer GCN for Euclidean models, 2-layer LGCN Chen et al. (2021) for FlatLand... For graph classification, we use 3-layer GIN Xu et al. (2018) as the Euclidean encoder, and the same 3-layer hyperbolic encoders as node classification for hyperbolic models" How LGCN is combined with the proposed method? What does the "3-layer hyperbolic encoders" mean in this case? I think the authors should first provide a clear explanation of these details in the main text before diving into the detailed results. Also, the paper can be reorganized to cover more important algorithmic details (i.e., pseudocode) in the main text.

- Lastly, I find that the authors provide the implementation of the method with a text link in the doc (line 404, "anonymous repository") instead of uploading it as a supplementary file. I nearly missed it. I would recommend the authors to upload the code directly whenever possible.

**Questions:**

1. Is the parameter $K$ in equation (4) just a learnable scalar? Do we need to have any kind of regularization/restrictions on $K$?

2. Line 501, what does "no parameter decoupling strategy" mean? Does it mean the server will aggregate all parameters including the client-learned curvatures?

---

> ### Author Response · Authors · 2024-11-21
>
> Thank you very much for your valuable comments and suggestions. Below, we will summarize the concerns with "**W:**" and our responses with "**R:**"
>
> ---
> **W1:** Typos.
>
> **R:** Thank you for your comments. We have corrected the identified typos and will conduct a thorough review to address any remaining issues.
>
> &nbsp;
>
> ---
>
> **W2 and Questions:** Ambiguous concepts.
>
> **R**: Thank you for your careful review. We appreciate your feedback and will address the ambiguity by adding clarifications to ensure the concepts are more precisely defined in the revised manuscript. Please let us address each point:
>
> **1. Regarding the operation function $ f(Wx, v) $ and the usage of $ v $:**
>
> The parameter $v$ is introduced to enable a more generalized formulation, supporting advanced non-linear operations with learnable parameters. While dropout is presented as a simple example without $v$ , the parameter can play a role in other minor operations within hyperbolic space. Specifically, $v$ can function as learnable parameters in normalization operations, constraining the norm via $ \sigma(v^T x) $ to prevent excessive growth of hyperbolic embeddings, or as bias terms added to $ x $. In later sections, the presentation is simplified to focus on the core linear transformation aspects (lines 283–284), as these additional operations do not influence the fundamental principles of our method.
>
> **2. Regarding Equation (4) and the usage of the first row of $ \hat{M}^{(l)} $:**
>
> The first row of $\hat{M}^{(l)}$ provides a generalized formulation for our defined LT operation, ensuring the correct input-output matrix size. The seemingly "missing" usage arises because our focus was on presenting the key transformations (lines 283–284). Typically, the first row’s parameters support minor operations. We will clarify this in the revised manuscript for better transparency.
>
> **3.Regarding $K$ and its implementation:**
>
> Yes, $ K $ is a learnable scalar parameter. To ensure the curvature remains negative (as required for hyperbolic space), we implement it as $ \text{sigmoid}(K) + 0.5 $. This design also keeps curvature $ - K $ within an effective range of $[0.5, 1.5]$, which prior work has shown to be an ideal range for most hyperbolic models [1-4]. Additionally, this approach maintains numerical stability while satisfying the need for a heterogeneous space.
>
> Thanks for bringing these points to our attention. We have revised the manuscript to include these clarifications in Appendix C2, making the technical details more transparent and accessible to readers.
>
> **4. Regarding no parameter decoupling strategy:**
>
> Yes. In the aggregation process, all parameters are aggregated including the curvature scalar ensuring consistency with our approach. We will clarify this in the revised manuscript.
>
> &nbsp;
>
> ---
>
> **W3:** Implementation of the graph classification task.
>
> **R:** Thank you for your comments. LGCN serves as the backbone for our graph learning framework, combining Lorentz linear layers (Equation 2) with graph aggregation operations, similar to how Euclidean counterparts like GCN and GIN integrate linear layers with graph aggregation. Each layer applies a Lorentz transformation followed by neighbor aggregation using the adjacency matrix to get the node representations. The "3-layer hyperbolic encoder" employs the same architecture with three stacked layers to learn node representations within each graph.
> For node classification, node representations are used directly, while graph classification employs mean pooling for graph-level representations.
>
> As noted in Section 5.3, the parameter decoupling strategy remains valid since "the aggregation operation does not involve any parameters."  This allows us to directly apply our methods to the parameters of the Lorentz linear layers without modification.
>
> Thank you for the suggestion to include more technical details in the main text. Due to time constraints and the page limit, moving more content to the main pages at this stage is challenging. *We will prioritize reorganizing the main text in the final version to better integrate these details.*
>
> &nbsp;
>
> ---
>
> **W4:** Submit code as supplementary materials.
>
> **R:** Thank you for your suggestion. We have now included our code as part of the supplementary materials for easier access and review.
>
> &nbsp;
>
> ---
> **References:**
>
> [1]  Fully hyperbolic neural networks. Chen, W., et al. (2021). ACL
>
> [2] Hyperbolic graph convolutional neural networks. Chami, I., et al. (2019). NeurIPS
>
> [3] HGCF: Hyperbolic graph convolution networks for collaborative filtering. Sun, J., et al. (2021). WWW
>
> [4]  Discrete-time temporal network embedding via implicit hierarchical learning in hyperbolic space. Yang, M., et al. (2021). KDD

---

> > ### Comment · Reviewer_4tQH · 2024-11-29
> >
> > I appreciate the authors' efforts for the detailed response and summary. I am more clear about the exp setup now. I personally do not hold objection for the paper. I think this work can inspire more ideas in federated hyperbolic learning problems.

---

> ### Author Response · Authors · 2024-11-29
> **Thank you!**
>
> Dear Reviewer 4tQH,
>
> &nbsp;
>
> Thank you very much for your positive feedback. We're glad our work resonated. Your comments motivate us to explore the potential of hyperbolic geometry in solving various challenging federated learning problems further, aiming to inspire more research in this emerging area.
>
> &nbsp;
>
> Best Regards,
>
> The Authors

---

### Official Review · Reviewer_nTmN · 2024-11-09

**Soundness:** 3
**Presentation:** 2
**Contribution:** 3
**Rating:** 6
**Confidence:** 3

**Summary:**

The paper proposes a new personalized federated learning algorithm called Flatland employing hyperbolic geometry. One of the key challenges in federated learning is the data heterogeneity among clients and understanding the similarity between clients data distributions can be helpful. The paper tries to achieve this goal by projecting each client data to a higher dimension to better capture similarities between clients data. The paper more focused on graph data. Experiments are carried out to showcase the effectiveness of the approach.

**Strengths:**

- The motivation and contribution of the paper is clear and it seems that leveraging hyperbolic geometric approaches can be helpful for personalized federated learning.
- The paper provides analysis in Section 6 to support their approach.
- Experimental study of the paper looks convincing.

**Weaknesses:**

Reading of the paper, I felt that some arguments in the paper are vague and needs more explanations to become more clear. For example "*PFL approaches include segmenting models into generic and personalized components (McMahan et al., 2017; Tan et al., 2023), leveraging model weights and gradients to map client relationships (Xie et al., 2021), or integrating additional modules to facilitate customization (Baek et al., 2023)*." Or for example in "Moreover, embedding data from various clients into a fixed space complicates the interpretability of model parameters, making it difficult to segment the model into meaningful components (Arivazhagan et al., 2019) and often expensive to assess similarities between client models", the reasons are not clear.

Furthermore, the paper states that embedding data from various clients into a fixed space often expensive to assess similarities between client models. However, using the Flatland itself in step 4 of algorithm 2, each clients needs to project its data to another space. I believe this can add more computations compared to other methods which does not employ Lorentz space and hyperbolic geometry-based approaches. Therefore, to me the above argument seems a bit contradictory.

**Questions:**

Looking into algorithm 2 of the paper, it seems that using Flatland clients and the server collaborate using conventional federated averaging methods such as Fedavg. Is Flatland compatible with other personalized federated learning algorithms? Is it possible to use other methods like PerFedAvg to orchestrate the collaboration between the server and clients? Maybe this can further improve personalization properties of Flatland.

---

> ### Author Response · Authors · 2024-11-21
> **Response**
>
> Thank you very much for your valuable comments and suggestions. Below, we will summarize the concerns with "**W:**" and our responses with "**R:**"
>
> ---
> &nbsp;
>
> **W1:** Some arguments in the paper are vague.
>
>
> **R:** Thank you for your comment. We will further refine our writing and try to provide as much explanation as possible within the limited space. Let us clarify these mentioned points:
>
> * The first passage highlights that PFL approaches address heterogeneity through three main strategies during aggregation: (1) splitting models into shared and personalized components, (2) analyzing weights or gradients to evaluate client similarities, and (3) incorporating additional modules for client-specific customization. Importantly, all these methods operate in Euclidean space.
>
> * The second passage emphasizes the challenge of embedding data from diverse clients into a fixed Euclidean space, as it complicates the interpretability of model parameters. In such settings, all parameters serve the same role during training, making it difficult to distinguish between those capturing client-specific heterogeneity and those representing shared patterns. This not only hinders meaningful model segmentation but also makes client similarity assessment more complex. Adding extra modules to address this further increases the system's complexity and reduces its flexibility.
>
> Thanks again for your comments. We have made modifications to the original text (lines 33–36 and lines 48–52) to address this ambiguity.
>
> &nbsp;
>
> ---
>
> **W2:** Additional operations from the Lorentz model.
>
>
> **R:** Thanks for your comment. We would like to clarify this contradiction:
>
> The computational complexity concerns raised in our original statement refer specifically to **assessing similarities between client models during the aggregation phase**, not the initial data projection. While FlatLand does indeed require projecting data into Lorentz space, this operation is computationally efficient for several reasons:
>
> 1. The exponential map is **a single input mapping function** (Equation 1) that depends only on the norm of the input:
> $\mathbf{x}^K = \left( \cosh \left(\frac{\||\mathbf{v}^E\||_2}{\sqrt{K}}\right), \sqrt{K} \sinh\left(\frac{\||\mathbf{v}^E\||_2}{\sqrt{K}}\right)\frac{\mathbf{v}^E}{\||\mathbf{v}^E\||_2} \right)
> $
>
>
> This projection has a constant time complexity of $O(1)$ per data point and can be **optimized through pre-computation** of input norms before training begins.
>
>
> 2. **No additional projections are required during training** after the initial mapping.
>
>
> 3. Most importantly, FlatLand **eliminates the need for expensive similarity computations during aggregation**. While other methods require computing pairwise model similarities or maintaining additional computational modules, our method performs a simple aggregation operation with complexity equivalent to standard FedAvg.
>
>
> 4. The effectiveness of FlatLand in **low-dimensional settings** (as demonstrated in Section 7.3) further reduces practical computational and communication costs.
>
>
> To make this clearer in the paper, we have added a detailed complexity analysis in Appendix B.6 that quantifies these trade-offs. The limited overhead from the initial projection is substantially outweighed by the computational savings during model aggregation and the reduced communication costs from effective low-dimensional representations.

---

> ### Author Response · Authors · 2024-12-01
>
> Dear Reviewer nTmN,
>
> &nbsp;
>
> We sincerely thank you for your positive feedback and efforts in helping us improve our paper! With only two days remaining before the deadline and having not yet received a response to our carefully formulated point-to-point replies, we kindly request your feedback. If you have any further questions or suggestions, we welcome the opportunity to discuss and address them.
>
> **We believe we have thoroughly addressed all the concerns raised in your initial review. And we would be grateful if you would consider raising your score accordingly.**
>
> Thank you once again for your efforts in ensuring the highest standards of academic excellence. **We look forward to your response and remain available for any further discussions.**
>
> &nbsp;
>
> Best Regards,
>
> The Authors

---

### Author Response · Authors · 2024-11-25
**Follow-up on Review Comments**

Dear Reviewers,

&nbsp;

Thank you very much for your valuable feedback and comments. As the rebuttal period is ending, please don't hesitate to reach out if you have any further questions or concerns. We would also appreciate it if you could update your ratings if we have resolved your concerns appropriately.

&nbsp;

Best regards,

Authors

---

### Author Response · Authors · 2024-12-03
**Rebuttal Summary**

&nbsp;

We thank all the reviewers for their time and for providing constructive comments to enhance the paper. **We appreciate the reviewers' positive comments:**

* The motivation and contribution of the paper are clear (*Reviews nTmN, 4tQH, Wtgh, Dc6W*). The ideas that integrating geometry information to solve the heterogeneity in federated learning are novel, interesting, and inspiring. (*All reviewers*)
* The proposed method is elegant (*Reviewer Dc6W*) , concise yet intuitive (*Reviewer 4tQH*), and easy to integrate. (*Reviewer KA7L*).
* Convincing evaluation and ablation study of proposed method in graph datasets. (*Reviewers nTmN, 4tQH, Wtgh, KA7L*)
* Solid theoretical support and sufficient analyses (*Reviewers nTmN, Dc6W*).
* This paper can inspire more ideas in federated hyperbolic learning problems (*Reviewer 4tQH, Dc6W*).

&nbsp;

---

We include the following revisions in the Rebuttal PDF according to reviewers' constructive suggestions to improve the quality of our manuscript. Due to the time constraints, we will continue to refine our paper later. Specifically, the added contents are as follows:

1. **Clearer arguments and typos.** We refined our explanations  (*Reviewers nTmN, 4tQHm*) and added experimental details about the curvature calculation  (*Reviewers 4tQH, Dc6W, 4tQH*) and graph classification settings in Appendix C2 (*Reviewer 4tQH*).

2. **Time complexity analysis.** We added a complexity analysis section in Appendix B6 to show that exponential maps do not lead to much extra computation overhead (*Reviewers nTmN, Dc6W*).

3. **Convergence analysis.** We added a convergence analysis in Appendix B5 to show that our method does not negatively impact the convergence rate (*Reviewer Dc6W*), which also enhances our theoretical analysis part (*Reviewers KA7L*).

4. **Experiments on image datasets.** We added a section in Appendix C3 to show some experiment results on the image dataset compared with some strong baselines that are not utilized in graph datasets (*Reviewers Dc6W, Wtgh, KA7L*).

5. **Experiments in a partial participation setting.** We added a section in Appendix C4 to analyze the performance with lower participation rate (*Reviewer KA7L*).

We update the whole results in Cora (50 clients) as follows; we will integrate and update it in our final version:


| Participate Rate |  0.1  |  0.3  |  0.5  |  0.7  | **1.0** |
| :--------------: | :---: | :---: | :---: | :---: | :-----: |
|      FedAvg      | 18.14 | 36.64 | 34.30 | 33.61 |  30.20  |
| FedPer       | 56.03 | 60.22 | 74.79 | 73.48 | 68.11 |
| FedHGCN      | 54.38 | 77.08 | 69.87 | 73.35 | 62.52 |
|   **FlatLand**   | **81.82** | **81.11** | **79.27** | **79.42** |  **77.98**  |

&nbsp;

---

We would like to give clarification in the question:

&nbsp;

**The selection of evaluation in graph datasets.**

The core idea of our approach lies in the correlation between data heterogeneity and hyperbolic curvature. We propose a novel method to leverage geometry information to decouple data heterogeneity. This relationship is particularly intuitive and supported by solid theoretical guarantees in graph data, where hyperbolic geometry has been more extensively developed. Therefore, we primarily use graph data to evaluate the rational and effectiveness of our proposed method.


&nbsp;

**Future work and improvement.**

We acknowledge that the performance of our method across all data types remains an open question for further investigation. Due to time constraints, our exploration of other data types is somehow limited, though we supplemented our study with experiments on image datasets. However, **this does not conflict with the motivation, contribution, methodology, and experimental results presented in this paper.** This work can inspire more ideas. Below, we outline possible areas for further exploration following this work:

1. It is the truth that hyperbolic space may NOT be universally optimal for all data distributions as it is still a rapidly evolving field. One intuition is that some data may exhibit positive curvature. Therefore, we could follow the idea of this paper to model more complex data structures in **mixed-curvature spaces** that also contain manifolds that have positive curvature.

2. Inspired by Reviewer KA7L, we could investigate the **unique potential** of geometric information when *dealing with the rapid updates of parameters for newly arrived users*. A novel strategy could be designed, for example, by *leveraging curvature values to directly identify clients with similar data properties*. This would allow us to quickly retrieve personalized parameters from these clients to help initialize the personalized parameters for the new user. This is a task that traditional Euclidean-space methods struggle with.

3. As hyperbolic space continues to show promising results across data in other domains, we could *expand our analysis and validation to include more complex and challenging datasets and tasks.*

---

### Meta-Review · Area_Chair_FZou · 2024-12-20

**Metareview:**

**(a) Summary of Scientific Claims and Findings:**
The innovations proposed by the authors are: i) use fully lorentz neural networks instead of just the hyperbolic graph neural networks, and ii) use client-specific curvatures. Their justifications are
- Fully Lorentz Neural Networks are even better than HGNNs at modeling hierarchical data due to the exponential volume expansion in hyperbolic space, enabling efficient representation of scale-free graphs with fewer dimensions than Euclidean models.

- Different clients may need different curvatures since the levels of complexity/hierarchy can be heterogenous. So the authors propose to learn personalized client-specific curvatures.

**(b) Strengths:**
- The idea of using client specific curvatures is well motivated and unique.
- The authors do a pretty convincing ablation study to demonstrate the effectiveness of their two innovations.

**(c) Weaknesses and Missing Elements:**
- Fully Lorentz Neural Networks are not novel - this is a well known technique in the centralized setting (Chen et al. 2022).
- While the results are great, the setup is currently quite limited. The number of clients and datasets are quite small. The effect of model architectures, client sampling, etc. is not explored.
- Theoretical contributions are limited and lack depth. Most the results are known or direct extensions.

**(d) Decision and Key Reasons for Rejection:**
The paper is rejected due to limited theoretical depth, lack of robust empirical validation across diverse datasets and realistic scenarios.

**Additional Comments On Reviewer Discussion:**

While the authors have made commendable efforts to address concerns during the rebuttal phase, key issues remain unresolved.

The theoretical contributions are limited, with the primary mathematical analysis (e.g., Corollary 1) being viewed as trivial or insufficiently impactful. The method’s theoretical motivation is noted but lacks rigorous exploration, such as convergence analysis or clear evidence linking Lorentz geometry to improved handling of heterogeneity across broader contexts. One reviewer expressed dissatisfaction with the explanation and validation of the parameter decoupling strategy, which is a central component of the proposed method. Specifically, the claim that decoupling separates heterogeneous and homogeneous information is not convincingly supported either theoretically or empirically.

**Suggestions for improvement from the reviews**

*Enhance Theoretical Depth*: Provide a more robust theoretical foundation, including convergence analysis, and explicitly demonstrate how Lorentz geometry uniquely models and mitigates client heterogeneity i.e. what are the kinds of heterogeneity can be uniquely captured by tuning the curvature.

*More diverse experiments*: Run larger real world experiment setups ideally on larger/harder datasets with a natural real world split. Also explore the effect of the model size. Can the hetergeneity across the clients be simply resolved by using larger models?

---

### Decision · Program_Chairs · 2025-01-22

Reject